



# A dense network of cosmic-ray neutron sensors for soil moisture observation in a pre-alpine headwater catchment in Germany

Benjamin Fersch[1], Till Francke[2], Maik Heistermann[2], Martin Schrön[3], Veronika Döpper[4], Jannis Jakobi[5], Gabriele Baroni[2,6], Theresa Blume[7], Heye Bogena[5], Christian Budach[2], Tobias Gränzig[4], Michael Förster[4], Andreas Güntner[7,2], Harrie-Jan Hendricks-Franssen[5], Mandy Kasner[3], Markus Köhli[8,9], Birgit Kleinschmit[4], Harald Kunstmann[1,10], Amol Patil[10], Daniel Rasche[7,2], Lena Scheiffele[2], Ulrich Schmidt[8], Sandra Szulc-Seyfried[2], Jannis Weimar[8], Steffen Zacharias[3], Marek Zreda[11], Bernd Heber[12], Ralf Kiese[1], Vladimir Mares[13], Hannes Mollenhauer[3], Ingo Völksch[1], and Sascha Oswald[2]

[1]Karlsruhe Institute of Technology, Campus Alpin (IMK-IFU), Kreuzeckbahnstraße 19, 82467 Garmisch-Partenkirchen, Germany
[2]Institute of Environmental Science and Geography, University of Potsdam, Karl-Liebknecht-Straße 24–25, 14476 Potsdam, Germany
[3]UFZ – Helmholtz Centre for Environmental Research GmbH, Dep. Monitoring and Exploration Technologies, Permoserstr. 15, 04318, Leipzig, Germany
[4]Technical University of Berlin, Geoinformation for Environmental Planning Lab, Straße des 17. Juni 135, 10623 Berlin, Germany
[5]Agrosphere IBG-3, Forschungszentrum Jülich GmbH, Leo-Brandt-Straße, 52425 Jülich, Germany
[6]Department of Agricultural and Food Sciences, University of Bologna, Viale Fanin 50, 40127 Bologna, Italy
[7]GFZ German Research Centre for Geosciences, Section Hydrology, Telegrafenberg, 14473 Potsdam, Germany
[8]Physikalisches Institut, Heidelberg University, Im Neuenheimer Feld 226, 69120 Heidelberg, Germany
[9]Physikalisches Institut, University of Bonn, Nussallee 12, 53115 Bonn, Germany
[10]Institute of Geography, University of Augsburg, Alter Postweg 118, 86159 Augsburg, Germany
[11]Department of Hydrology and Water Resources, University of Arizona, 1133 E. James E. Rogers Way, 85721-0011 Tucson, Arizona, USA
[12]Institute of Experimental and Applied Physics, University of Kiel, German
[13]Helmholtz Zentrum München, Institute of Radiation Medicine, Neuherberg, Germany

**Correspondence:** Benjamin Fersch (fersch@kit.edu)

**Abstract.**

Monitoring soil moisture is still a challenge: it varies strongly in space and time and at various scales while well established sensors typically suffer from a small spatial support. With a sensor footprint up to several hectares, Cosmic-Ray Neutron Sensing (CRNS) is an emerging technology to address that challenge.

So far, the CRNS method has typically been applied with single sensors or in sparse national scale networks. This study presents, for the first time, a dense network of 24 CRNS stations that covered, from May to July 2019, an area of just 1 km$^2$: the pre-alpine Rott headwater catchment in Southern Germany which is characterized by strong soil moisture gradients in a heterogeneous landscape with forests and grasslands. With substantially overlapping sensor footprints, that network was designed to study root zone soil moisture dynamics at the catchment-scale. The observations of the dense CRNS network were

complemented by extensive measurements that allow to study soil moisture variability at various spatial scales: roving (mobile)



CRNS units, remotely sensed thermal images from Unmanned Areal Systems (UAS), permanent and temporary wireless sensor networks, profile probes as well as comprehensive manual soil sampling. Since neutron counts are also affected by hydrogen pools other than soil moisture, vegetation biomass was monitored in forest and grassland patches, as well as meteorological variables; discharge and groundwater tables were recorded to support hydrological modeling experiments.

As a result, we provide a unique and comprehensive data set to several research communities: to those who investigate the retrieval of soil moisture from cosmic-ray neutron sensing, to those who study the variability of soil moisture at different spatio-temporal scales, and to those who intend to better understand the role of root-zone soil moisture dynamics in the context of catchment and groundwater hydrology, as well as land – atmosphere exchange processes. The data set is available through EU-DAT, splitted into the two subsets https://doi.org/10.23728/b2share.85fe0f9dac0f48df9215c17e65d1f1e1(Fersch et al., 2020a)

and https://doi.org/10.23728/b2share.93ed99e486904d48a8a6a68083066198 (Fersch et al., 2020b).

# 1   Introduction

## 1.1   The relevance of soil moisture observation

Soil moisture is a key state variable of the Earth's environmental system, controlling various processes at various scales: the

exchange of water and energy between the land surface and the atmosphere, runoff generation and groundwater recharge, vegetation development and growth in natural and managed systems, or the release of greenhouse gases from soils.

    But while soil moisture is a much desired quantity in research and applications, its observation remains a challenge (Beven et al., 2020; Ochsner et al., 2013). Numerous techniques exist for measuring soil moisture at specific points in space, including vertical profiles, such as TDR, FDR, and soil sampling. While these techniques all have specific uncertainties and limitations –

including even the benchmark standard technique of thermo-gravimetry – they share one fundamental limitation: the uncertain spatial representativeness. This shortcoming results from the small support (footprint of the measurement in the order of centimeters) in combination with the strong and potentially abrupt variation of soil moisture in space. This issue can be overcome by very dense in-situ sensor networks (see, e.g., Bogena et al., 2010). At the other end of the scale continuum, remote sensing techniques allow for a higher spatial coverage and a volume- or area-integrated measurement approach. Wang

and Qu (2009) and Mohanty et al. (2017) provide an overview of techniques based on optical, thermal, passive microwave, and active microwave measurements. Yet, apart from the fact that these are indirect observations and thus inherently uncertain, they typically have limited (and often imprecisely characterized) penetration depths that cannot be assumed to be representative for the root zone. Over densely vegetated terrain the uncertainty of these remote sensing techniques increases. Furthermore, both airborne and spaceborne remote sensing of soil moisture often come along with infrequent measurement intervals due to a

limited overpass frequency.





## 1.2 Cosmic ray neutron sensing of soil moisture

Cosmic-Ray Neutron Sensing (CRNS) is a promising measurement technique to close the scale gap in soil moisture observation. According to simulations by Köhli et al. (2015), the sensor can provide volume-integrated soil moisture estimates with an exponentially-shaped horizontal footprint of hundreds of meters (the "plot scale") and a vertical footprint of tens of centimeters, corresponding to the upper root zone. Bogena et al. (2013) and Schrön et al. (2018) showed that the temporal resolution of standard detectors is in the range of 3 to 12 hours and strongly depends on the detector technology used. The principles of the CRNS method were introduced about twelve years ago by Zreda et al. (2008), followed by Desilets et al. (2010), who presented first results using mobile measurements and a conversion procedure to obtain soil moisture. Zreda et al. (2012) presented a complete overview over the method and the newly established CRNS network in the USA. Andreasen et al. (2017) published a short synthesis of recent advancements in CRNS at that time, but soon afterwards important updates with respect to snow detection (Schattan et al., 2017) and road effects on roving CRNS measurements (Schrön et al., 2018) were published.

The CRNS sensor is sensitive to the ambient density of neutrons in the near-surface atmosphere. The main source of cosmic-ray induced neutrons exhibits energies between $10^5$ and $10^9$ eV (Köhli, 2019). Their interaction in the ground leads to neutrons in the energy range of $10^{-2}$ to $10^7$ eV, of which epithermal neutrons ($1$–$10^5$ eV) are most sensitive to hydrogen and water (Zreda et al., 2012; Köhli et al., 2015). Hence, standard CRNS sensors are equipped with a polyethylene shield that reduces the thermal neutron ($< 1\,$eV) fraction and slows down faster neutrons to detectable energies (Köhli et al., 2018). Some CRNS sensors employ an additional "bare" counter which is sensitive to thermal neutrons only.

To date, a wide range of neutron detectors optimized for soil moisture monitoring exists, using various gases or coating materials for neutron detection. An overview of detectors used in our field campaign is provided in section 3.4. For a detailed description of the technical components that make up standard CRNS units, including detector, telemetry, and additional atmospheric sensors, the reader is referred to Zreda et al. (2012) and Schrön et al. (2018).

The intensity of detected neutrons is mainly controlled by the interaction with hydrogen pools in the sensor footprint, of which soil moisture is typically the most important, though not the only one. A standard approach to estimate the gravimetric ($\theta_{\mathrm{grv}}$) or volumetric soil moisture ($\theta_{\mathrm{vol}}$, often referred to as $\theta$) from epithermal neutron count rates $N$ is to use the transfer function proposed by Desilets et al. (2010):

$$\theta_{\mathrm{grv}} = \frac{a_0}{N/N_0 - a_1} - a_2\,, \quad \theta_{\mathrm{vol}} = \frac{\varrho_b}{\varrho_w} \cdot \theta_{\mathrm{grv}} \tag{1}$$

In that form, the transfer function requires the calibration of parameter $N_0$ once, in order to take the specific measurement conditions into account. The value not only depends on the local terrain features, soil moisture heterogeneity, and instrumental sensitivity (Schrön et al., 2018), but also on the local prevalence of other hydrogen pools, such as vegetation, litter, soil organic carbon, and lattice water (Andreasen et al., 2017). The shape parameters $a_0$, $a_1$, and $a_2$ could be adapted to specific local conditions, but they have also proven to be robust in many previous studies. Soil bulk density $\varrho_b$ (kg/m$^3$) needs to be measured or estimated locally, while the density of water, $\varrho_w$, can be assumed to be $1000\,$kg/m$^3$.





The calibration of $N_0$ requires the availability of soil moisture observations within the horizontal and vertical footprint of the sensor. Schrön et al. (2017) have synthesized the state-of-the-art methodology on how to make use of distributed point-scale soil moisture measurements for sensor calibration and proposed adequate sampling designs.

Yet, calibration can only account for the effect of static hydrogen pools at a specific point in time, while hydrogen pools such as vegetation are typically dynamic. In addition, the collection and processing of soil samples for measuring thermo-gravimetric soil moisture is particularly labour-intensive.

One of the challenges in using CRNS for soil moisture observation is thus to separate the part of the signal that is related to soil moisture from those parts affected by other hydrogen pools such as vegetation and soil organic carbon (Baatz et al., 2015; Andreasen et al., 2017; Jakobi et al., 2018). Further prerequisites include the correction of atmospheric effects such as air pressure, air humidity, and fluctuations of incoming neutron radiation (Rosolem et al., 2013; Schrön et al., 2016; Hawdon et al., 2014), and a better, physically-based understanding of the spatial sensitivity of the neutron sensors, both vertically and horizontally, and how it is influenced by dynamic environmental conditions in the footprint (Köhli et al., 2015; Schrön et al., 2017; Schattan et al., 2019).

While the horizontal footprint of a single CRNS sensor already exceeds the spatial support of conventional techniques, further attempts have been made to enhance the spatial coverage: *roving CRNS* and *networks of CRNS sensors*. *Roving CRNS* involves mobile CRNS sensors, have been recognized as a promising approach to increasing the spatial scale (see, e.g., Chrisman et al., 2013). Within a CRNS roving campaign, the sensor unit is moved (e.g., by car) within the area of interest in order to explore the catchment-scale wetness conditions and related spatial patterns (Dong et al., 2014; Franz et al., 2015; McJannet et al., 2017; Schrön et al., 2018). Under ideal situations, up to hundreds of square kilometers may be covered in a single day. This coverage makes CRNS roving a promising method for closing the critical scale gap in soil moisture monitoring toward the scale of small to medium catchments. However, roving relies on campaign-based measurements and thus produces snapshots only. As roving requires neutron measurements with much higher temporal resolution (in the order of minutes), a CRNS rover usually consists of several larger detectors, resulting in an increase in achievable neutron count rates compared to stationary applications. In this context, an increase of detector sensitivity would constitute a promising perspective: obviously, the potential to increase the signal-to-noise ratio by integrating neutron counts over time is limited if the sensor is mobile.

In contrast, *networks of CRNS sensors* operate multiple stationary CRNS-sensors. Some initiatives have established national CRNS monitoring networks with the aim of supporting environmental monitoring at larger scales. These networks are being implemented under the acronym of COSMOS (the COsmic-ray Soil Moisture Observing System). The first network was established in the USA by the University of Arizona and has already deployed more than 60 CRNS sensors at various locations across the USA (Zreda et al., 2012). The Australian network was supported by the CSIRO research institute and consists of nine sensors distributed across the continent under different environmental conditions (Hawdon et al., 2014). A network has also been established in the United Kingdom by the CEH  (Evans et al., 2016). Similar initiatives have been started in Kenya and India (Montzka et al., 2017). Yet, these networks aim at distributing single CRNS sensors at a national scale, similar to the idea of climate stations, providing localized measurements in a sparse network that spans large scales.





To better cover the spatial and temporal scales, the combination of both stationary and roving approaches has been explored (Chrisman and Zreda, 2013; Franz et al., 2015): a limited number of stationary sensors have been used to detect the temporal dynamics while roving surveys have provided the means for extra- and interpolation beyond the stationary sensors. Despite the
promising results, many challenges have been identified due to the heterogeneity of land surface conditions and the validity of assumptions that govern the integration of roving CRNS data, which are sparse in time, with stationary CRNS data, which are sparse in space.

### 1.3 The Cosmic Sense Joint Field Campaign in 2019

The Cosmic Sense Research Unit, funded by the German Research Foundation (DFG), addresses the above challenges in a
concerted effort with a consortium of eight German partner institutions. In this context, one component of the Cosmic Sense project is the targeted joint operation of a large number of CRNS sensors in a dense observational network. The scientific aims behind these "Joint Field Campaigns" (JFCs) are

- to systematically explore, at the landscape level, the effect of heterogeneity of soil and vegetation in the CRNS footprint on the neutron signals;

- to investigate the consistency of signals obtained from CRNS sensors of different manufacturers and sensitivities;

- to monitor water storage in the root zone, as the most dynamic storage component in a catchment, and to possibly assimilate these observations in a hydrological model;

- to establish a network with overlapping horizontal CRNS footprints in order to investigate the potential of constraining soil moisture estimates in space and time and thus to establish space-time representations of soil moisture at a resolution
higher than the resolution of a single sensor;

- to better understand the influence of hydrogen pools other than soil moisture on the CRNS signal, and to develop corresponding correction procedures;

- to evaluate the relation between the spatial and temporal dynamics of CRNS footprints and thermal infrared remote sensing imagery.

As mentioned in the previous section, networks of a large number of CRNS sensors had been established before; however, as sparse national scale networks they are not geared towards the scientific subjects elaborated above. Hence, the joint operation of 24 CRNS sensors that took place from early May to late July 2019, is unprecedented in its scope as it features a large number of detectors in an area of just $1\,\mathrm{km}^2$, characterised by pronounced soil moisture gradients and heterogeneous land cover. The CRNS monitoring network was complemented by various observational techniques including CRNS roving, thermal
imaging from Unmanned Aerial Systems (UAS), multiple FDR-sensor networks (SoilNet clusters) and profile probes, as well as manual measurements with FDR probes and thermo-gravimetry; and, finally, embedded in an established long-term observation infrastructure (TERENO Zacharias et al., 2011) and multiple synchronous observational campaigns (MOSES, ScaleX, see





Sect. 4.1) that focused on water and energy fluxes in the atmospheric boundary layer. The observational period from late spring
to midsummer was characterized by pronounced soil moisture dynamics, starting out from fully saturated conditions after
persistent heavy rainfall, followed by more than two months of drying, occasionally interrupted by brief but intense rainfall
events.

In this paper, we present this unique data set as a contribution to the CRNS community, but also to those researchers interested
in exploring the potential of dense volume-integrated soil moisture observations from a hydrological perspective. The data set
was split into two subsets and is freely available from EUDAT at (https://doi.org/10.23728/b2share.85fe0f9dac0f48df9215c17e65d1f1e1,
Fersch et al., 2020a) and (https://doi.org/10.23728/b2share.93ed99e486904d48a8a6a68083066198, Fersch et al., 2020b).

We start by providing a detailed overview and justification of the choice of the study area (section 2). Section 3 constitutes
the core of this paper: the presentation of the data collected as part of the JFC. Section 3.1 gives an overview of the various
components, section 3.2 documents the underlying data model, and the subsequent sections describe these data sets in more
detail. Section  4 outlines relevant data from third parties which however where not collected as part of the JFC, and are not
part of this data publication. As this paper is about the presentation of the data set, we will not provide any further analysis
or interpretation. However, in section 5, we will illustrate how the neutron count rates of stationary and roving sensors can be
used to represent temporal soil moisture dynamics at the catchment scale, including the areal average of soil moisture and its
variability in space. Based on that example, section 6 will conclude by outlining the potential of the presented data set.

## 2   Study site

In the selection process for the field campaign's study location the requirements of the different involved research projects
needed to be reconciled. Important criteria were backbone climate and soil-moisture observations, contiguous landcover, good
accessibility by foot and car for the mobile applications, a substantial fraction of nonforested area for the remote sensing
(UAS) campaigns, shallow and time-variable groundwater levels, and a self-contained hydrological catchment to enable hy-
drological modeling. With the $1 \, \mathrm{km^2}$ Rott headwater catchment (Fig. 1), part of the Pre-Alpine Terrestrial Environmental
Observatory (TERENO, Kiese et al., 2018) of the Helmholtz association, we could identify a suitable candidate. TERENO
Pre-Alpine is situated about $50 \, \mathrm{km}$ southwest of Munich, Bavaria, Germany and encompasses the $600 \, \mathrm{km^2}$ Ammer and the
$55 \, \mathrm{km^2}$ Rott watersheds.

The Fendt site of TERENO Pre-Alpine ($595 \, \mathrm{m}$ ASL) is located in the Rott headwater catchment. In the area, several
multi- and interdisciplinary campaigns as well as some long-term ecological experiments were being carried out over the
past years (e.g., SCALEX, Wolf et al., 2016; Kiese et al., 2018). Time-limited campaigns such as the JFC of Cosmic Sense
can help to complement the long-term observations at the TERENO Pre-Alpine locations, whereas on the other hand they can
build on preexisting knowledge and measurements.

The younger morphodynamics of the region were governed by glacial and post-glacial processes of the Quaternary. The
shallow, U-shaped valley of the Rott was carved into older sediments (molasse) by the Isar-Loisach glacier about $25 \, \mathrm{k}$ years
ago and followed by kettle lake sedimentation and fluvial erosion processes. Whereas towards the side slopes of the valley



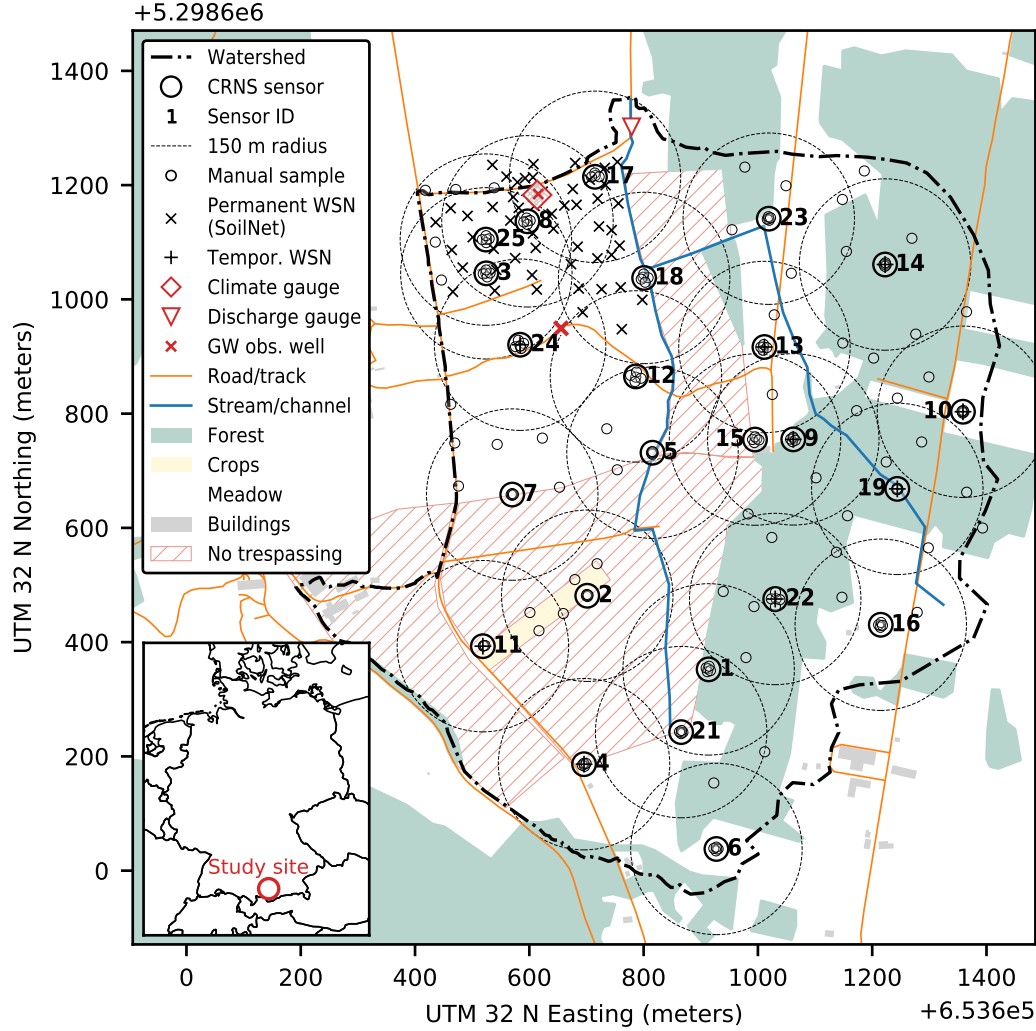

**Figure 1.** The Rott headwater catchment in Fendt near Peißenberg, Bavaria, Germany: the map shows the most important features of the study area and the JFC. In terms of land use, we only highlight forested area and a patch of cropland – the remaining parts are mainly meadows or grassland, with some scattered settlements – as well as roads and streams. In terms of instrumentation, we focus on showing the locations of the CRNS sensors with a 150 m radius as a typical footprint size for a medium level of soil moisture (Schrön et al., 2017), the climate station, and soil moisture profiles from SoilNet nodes and manual sampling on June 25–26, 2019. Basemap data from OSM (© OpenStreetMap contributors, 2019), see section 4.4.

gravels mixed with loamy and silty fractions are predominant, we find mainly silty and loamy sediments at the lower elevations with peaty compositions towards the draining rivulet (Fersch et al., 2018).



In the Rott headwater catchment, between the deeper Tertiary (molasse) and the Quaternary sedimentation layers, shallow aquifers have formed with hydraulic heads that range between 4 and 0.2 m below ground from the margins to the center of the headwater catchment. The prevalent soil class is Cambic Stagnosol, which originated from the glacial parent material. Typical clay, silt, and sand fractions are 32, 41, and 27 % (Kiese et al., 2018). We present a description of soil properties including bulk density, lattice water content, organic matter and texture for composites in subsection 3.8. As can be seen from Fig. 1, the landcover of the headwater catchment consists mainly of grassland, with some wetter regions along the creeks of the northern part, and a forest on the eastern slope consisting of stands of coniferous or deciduous species with heterogeneous age.

The boundaries of the Rott headwater catchment were delineated according to the surface topography (DEM1 1 m x 1 m, https://www.ldbv.bayern.de). For the subsurface (groundwater) we assume additional contributions from the adjacent hillslopes to the west.

## 3   Methods and data

### 3.1   Overview

This section describes the measurements conducted during the campaign. Other complementary data are described in section 4.

The core innovation and overall motivation of this data set is the use of a dense network of 24 CRNS sensors in a study area of roughly 1 km$^2$. These instruments of various types and their key observational variables are described in Tab. 2.

Around this core data set we carried out various measurements that are required to utilize, study and evaluate observed neutron counts from the CRNS network for the purpose of soil moisture retrieval: meteorological observations (section 3.3); soil properties and soil moisture measurements for calibration and validation, obtained at selected points and various depths using different techniques (section 3.8); CRNS roving data (section 3.6) and remotely sensed thermal images (section 3.10), both from several campaigns in order to obtain additional information on soil moisture variability in space; and measurements and estimates of above-ground biomass (section 3.9) in order to account for the corresponding hydrogen pools in the CRNS data analysis.

### 3.2   Data formats

The data presented in this study consist, for large parts, of time series data recorded at well-defined, sparse and static points in space (e.g., neutron counts, meteorological variables, soil moisture or permittivity). For such data, we decided to implement a simple, transparent, and easy-to-use data model that is based on standard text tables (character (tabulator) separated values, csv): each sensor (or sensor unit) is characterized in attribute tables, including a unique identifier (ID), location (latitude and longitude in WGS 84 reference system), measurement depth or height (in meters above or below the surface), and a set of sensor specific attributes that are documented in the attribute table's header. The time series of observations are provided in additional text tables: each sensor is represented by one file that is named according to the unique sensor ID. The first column in





any such file holds the datetime (UTC, ISO 8601). Any other columns represents the measured variables which are documented in the file header, including the physical units.

Timeseries data with a fixed spatio-temporal resolution (e.g., data from SoilNet) are provided in NetCDF files which include an explicit documentation of all dimensions and observational variables, but which are also accompanied by an overview text table that contains key attributes of the measurement locations.

Any exceptions from these data models (e.g., for the CRNS roving, the land use and soil data, the digital elevation model, or the remote sensing data) are explicitly elaborated in the corresponding subsections, and by using meta data files. For polygon

data, we use the ESRI shapefile and the geojson format, for gridded data the GeoTIFF format.

Please see details of the data repository in section 7.

### 3.3 Meteorological data

The permanent backbone meteorological instrumentation at the Rott headwater catchment (also known as the TERENO site Fendt) consists of an eddy-covariance flux tower and several precipitation sensors. Besides the high frequency measurements of

the eddy-flux system (the data of which is available through the Integrated Carbon Observation System ICOS and TERENO), standard climate variables are recorded every minute. The meteorological observations and their respective devices selected for the presented data set are listed in Tab. 1.

**Table 1.** Meteorological instrumentation in the Rott headwater catchment (instrumentation is part of the TERENO Pre-Alpine Observatory at the Fendt site).

| Variable | Sensor Brand | Accuracy | Precision |
|---|---|---|---|
| air temperature | WXT520, Vaisala | ± 0.3 °C | 0.1 °C |
| air rel. humidity | WXT520, Vaisala | ± 3 % | 0.1 % |
| air pressure | WXT520, Vaisala | ± 5 mbar | 1 mbar |
| global radiation | SPN1, DeltaT Devices | ± 8 % | 0.6 W m$^2$ |
| precipitation | Pluvio$^2$, Ott Hydromet | ± 1 % | 0.01 mm |
| wind speed | WXT520, Vaisala | ± 3 % | 0.1 m s$^{-1}$ |
| wind direction | WXT520, Vaisala | ± 3° | 0.1° |

### 3.4 Stationary CRNS data

During the period from mid-May to mid-July 2019, a total of 24 stationary CRNS sensors were operated, though not all sensors

were measuring at all times. CRNS sensor #8 in the north-east had already been installed as part of TERENO infrastructure, and continued operation after the JFC. The temporal data availability is illustrated in Fig. 9, center panel. Data gaps and differing periods of data availability are due to various reasons, including different dates for installation/deinstallation, sensor maintenance, power failures, or logger configuration, to name a few.

Tab. 2 gives an overview of all stationary CRNS sensors. Out of the 24 sensors, 18 have been manufactured by Hydroinnova
LLC (Albuquerque, NM, USA). Those instruments are based on neutron-sensitive detector gases, which are either $^3$He gas
(CRS-1000, CRS-2000) or $^{10}$BF$_3$ enriched gas (CRS-1000-B, CRS-2000-B). Five sensors (StX-140-5-15) were manufactured
by StyX Neutronica GmbH (Dossenheim, Germany), and feature a new experimental design, based on solid $^{10}$B-lined convert-
ers. One of the 24 sensors was manufactured by Lab-C LLC (Tucson, AZ, USA), and also constitutes a new approach that uses
a multi-wire proportional chamber with solid $^6$Li as a neutron converter. Fig. 3 exemplifies time series of raw neutron count
rates from six CRNS sensors of different types (manufacturers) and thus different sensitivities.

For all CRNS sensors, the detection chamber is surrounded by a so-called moderator material in order to "thermalize" (i.e.
slow down) epithermal neutrons for detection (see Zreda et al., 2012; Schrön et al., 2018; Köhli et al., 2018, for details). Six
sensor units were equipped with an additional detection unit without a moderator (i.e. a "bare counter") in order to directly
count thermal neutrons (see Tab. 2). This approach was motivated by recent studies which show that the ratio of thermal to
epithermal neutron count rates is useful to distinguish specific hydrogen pools, namely vegetation biomass (Tian et al., 2016;
Jakobi et al., 2018) or snow (Schattan et al., 2017).

The CRNS sensor units were equipped with varying meteorological sensors for air temperature, relative humidity and air
pressure, all of which are required to correct for atmospheric effects on epithermal neutron count rates. For some units, such
sensors were only available internally in the logger box (making the observations less representative for the sensor footprint),
while some units also featured external meteorological sensors. A more detailed specification of the placement of meteorolog-
ical sensors is included in the metadata of the individual CRNS sensor units. All CRNS sensors were set up to record neutron
counts and additional variables at a temporal interval of 20 minutes.

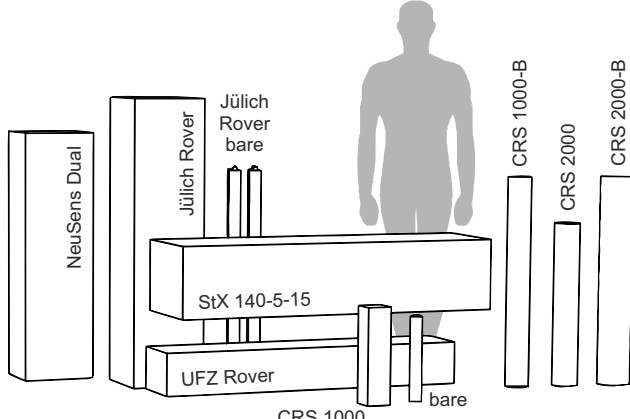

**Figure 2.** Dimension of CRNS sensors employed in the campaign. The blocks illustrate the size of the detectors; the actual units also
comprise other components, combined in a slightly larger housing.

At 19 of the 24 CRNS sensor locations a profile probe (40 or 100 cm maximum measurement depth, see section 3.8.3 for
further details) was installed to monitor the vertical distribution of soil moisture.

**Table 2.** Overview of CRNS sensors installed during the JFC, including: manufacturer, model, and converter technology; the availability of standard moderated detector tubes for epithermal neutrons (mod) and additional bare tubes for thermal neutron detection (bare); the dominant land cover in the footprint of the sensor; the maximum measurement depth of co-located profile probes near the CRNS sensor; the sensitivity factor (ratio between raw neutron counts of the individual sensor and the calibrator unit (#20).

| ID | Manufacturer | Sensor model | Technology | Tubes | Dominant land cover | Profile depth | Sensitivity |
|----|-------------|--------------|------------|-------|--------------------|--------------| ----------|
| 1 | Hydroinnova | CRS 2000-B | $^{10}BF_3$ | mod & bare | forest, meadow | 100 cm | 1.190 |
| 2 | Hydroinnova | CRS 1000 | $^{3}He$ | mod | crops, meadow | 100 cm | 0.452 |
| 3 | Hydroinnova | CRS 1000 | $^{3}He$ | mod & bare | meadow | – | 0.458 |
| 4 | Lab-C | NeuSens dual | $^{6}Li$ | mod | meadow | 100 cm | 4.530 |
| 5 | Hydroinnova | CRS 1000-B | $^{10}BF_3$ | mod | meadow | 100 cm | 0.670 |
| 6 | Hydroinnova | CRS 1000-B | $^{10}BF_3$ | mod | meadow, forest | 100 cm | 0.668[*] |
| 7 | Hydroinnova | CRS 1000-B | $^{10}BF_3$ | mod | meadow | 100 cm | 0.668[*] |
| 8 | Hydroinnova | CRS 2000-B | $^{10}BF_3$ | mod & bare | meadow | – | 1.161 |
| 9 | StyX Neutronica | StX-140-5-15 | $^{10}B$ | mod | forest, meadow | 100 cm | – |
| 10 | StyX Neutronica | StX-140-5-15 | $^{10}B$ | mod | meadow, forest | 100 cm | – |
| 11 | StyX Neutronica | StX-140-5-15 | $^{10}B$ | mod | crops, meadow | 100 cm | – |
| 12 | StyX Neutronica | StX-140-5-15 | $^{10}B$ | mod | meadow | 100 cm | – |
| 13 | StyX Neutronica | StX-140-5-15 | $^{10}B$ | mod | meadow | 40 cm | 0.984 |
| 14 | Hydroinnova | CRS 2000 | $^{3}He$ | mod | forest | 100 cm | 0.871 |
| 15 | Hydroinnova | CRS 2000 | $^{3}He$ | mod | meadow, forest | 100 cm | 0.871[*] |
| 16 | Hydroinnova | CRS 2000-B | $^{10}BF_3$ | mod & bare | meadow | 40 cm | 1.148 |
| 17 | Hydroinnova | CRS 2000-B | $^{10}BF_3$ | mod & bare | meadow | – | 1.121 |
| 18 | Hydroinnova | CRS 1000 | $^{3}He$ | mod & bare | meadow | 100 cm | 0.414 |
| 19 | Hydroinnova | CRS 2000-B | $^{10}BF_3$ | mod | forest | 40 cm | 1.147[*] |
| 20 | Hydroinnova | Calibrator | $^{3}He$ | mod | – | – | 1.000 |
| 21 | Hydroinnova | CRS 2000-B | $^{10}BF_3$ | mod | meadow, forest | 100 cm | 1.132 |
| 22 | Hydroinnova | CRS 2000-B | $^{10}BF_3$ | mod | forest | 40 cm | 1.168 |
| 23 | Hydroinnova | CRS 2000-B | $^{10}BF_3$ | mod | meadow, forest | 40 cm | 1.127 |
| 24 | Hydroinnova | CRS 2000-B | $^{10}BF_3$ | mod | meadow | 100 cm | 1.138 |
| 25 | Hydroinnova | CRS 1000-B | $^{10}BF_3$ | mod | meadow | – | 0.665 |

[*] No direct measurement available, values were obtained by using the average of same sensor models as a reference.

The locations of the CRNS sensors are shown in Fig. 1. The sampling design underlying the placement of sensor units was subject to various scientific and practical – partly antagonizing – constraints:

– to achieve a maximum coverage of the catchment area in order to capture soil moisture storage dynamics in the root zone as an important dynamic part of the catchment's water balance and a key control of groundwater recharge;



- – antagonistic to the former, to achieve a maximum overlap of CRNS footprints in order to better constrain the estimation of soil moisture patterns in space and time from multiple CRNS signals;

- – to achieve a balance of coverage between the different land cover and soil types, most notably with regard to 'meadow' versus 'forest' cover and 'loamy/silty' versus 'peaty' soils;

- – to represent different positions along hill slopes inclined towards the northward draining rivulet;

- – to avoid properties for which owners did not grant permission for trespassing nor installing equipment (mostly in the West-Southwest);

- – to use locations with sufficient insolation for solar panels;

- – to efficiently incorporate existing observational infrastructure (such as the SoilNet in the North-West);

- – to keep a minimum distance of 15 meters to roads in order to minimize the effect of roads on CRNS measurements (Schrön et al., 2018);

- – to ensure proximity to tracks for the installation of the heavy StX-140-5-15 units and to enable comparisons with the cosmic rover measurements.

The trade-off between the aim of maximum coverage versus maximum overlap was resolved by increasing the density of CRNS sensors in the north west, where the permanent SoilNet allows for optimal validation.

## 3.5 Standardisation of neutron count rates

Neutron count rates at different observation locations were obtained with different CRNS sensor types. Even within the same sensor type, the effective sensitivity varies. Between all sensors used in this study, the sensitivity between the least and the most sensitive sensor (CRS-1000 and Lab-C, respectively) is expected to vary by an order of magnitude. In order to compare neutron count rates observed at different locations, these count rates need to be normalized to a standard level. Given that the effective sensitivity of the instrument is unknown, we need to introduce a reference standard. For that purpose, we placed a mobile CRNS sensor ("calibrator", Hydroinnova, #20 in Tab. 2, basically consisting of two combined CRS-1000 systems) just beside each stationary CRNS sensor for a period of at least 24 hours. The ratio between the average count rates of the stationary CRNS sensor and the calibrator - the sensitivity factor - was used to standardize the count rates for the entire time series to the calibrator level. For those CRNS sensors which we could not collocate with the calibrator, we used the average sensitivity factors obtained for the same instrument type. The resulting sensitivity factors are included in Tab. 2. Furthermore, a CRNS rover unit was collocated with most of the stationary CRNS sensors on June 27 which provides a further sensitivity reference (see section 3.6).





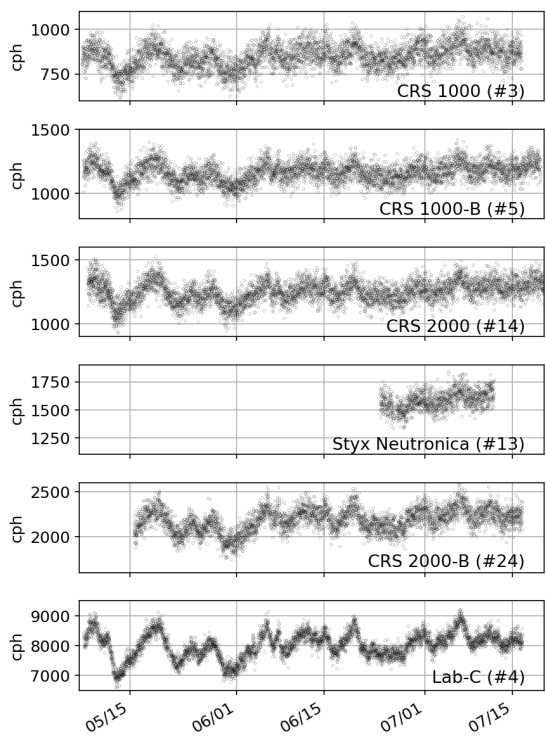

**Figure 3.** Raw (not standardized, uncorrected) neutron count rates (in counts per hour, cph) recorded at an integration interval of 20 minutes by six exemplary CRNS sensors representing different manufacturers/models (sensor ID in parentheses, see Tab. 2). While the qualitative dynamics are visually similar, the different sensitivities become evident by both different averages and different signal-to-noise ratios - please mind the different y-axis scaling.

### 3.6 Roving CRNS

During the campaign, portable cosmic-ray neutron sensors were used:

- to study the spatial variability of neutron intensity and soil moisture in the whole catchment and particularly in-between the stationary sensors;

- to validate the spatial representativeness of the stationary sensors and its dependency on different land use types;

- to investigate differences in the sensitivity of the stationary sensors by using the mobile sensor as a reference standard.

We used three types of mobile devices:

- UFZ rover (installed in a Land Rover Defender) equipped with a moderated CRNS-RV unit (Hydroinnova LLC, Albuquerque, USA) based on $^3$He gas (see Schrön et al., 2018, for details). In contrast to previous studies, we applied an additional polyethylene shield of 5 cm thickness below the detector to reduce local effects. Air temperature and humidity



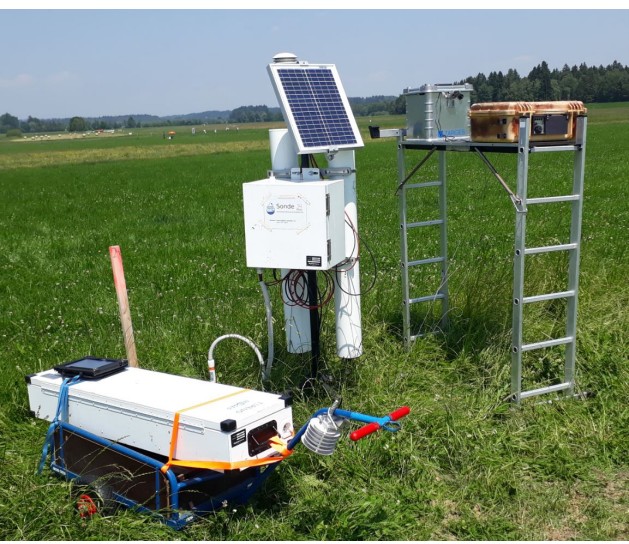

**Figure 4.** Inter-calibration of the stationary sensor (middle) using the mobile rover on a hand wagon (left) and the mobile calibrator unit (right), with minimum record periods of 20 min and 24 h, respectively.

were recorded with sensors attached outside the car. Three consecutive measurements underwent a moving-average filter to account for the moving footprint and to reduce the relative statistical uncertainty. The average epithermal neutron count rate across all campaigns was $9\,788 \pm 2\,141$ cph prior to corrections;

– hand wagon equipped with the identical detector system as in the UFZ rover (Fig. 4). This type of measurement has been applied mainly on June 27[th] for sensor intercalibration and to cross terrain that is inaccessible for cars. On a few days in May and June, the same measurement principle has been used for further sensor intercalibration;

   – FZJ rover (Mercedes Sprinter) equipped with an array of nine neutron detector units (Hydroinnova LLC, Albuquerque, USA), each holding four tubes filled with $^{10}BF_3$. During the campaigns, the rover was configured to measure epithermal

neutrons with five detector units, three in vertical and two in horizontal orientation. The remaining four bare detector units measured thermal neutrons during the experiments to calculate the thermal-to-epithermal neutron ratio $N_r$ (see, e.g., Tian et al., 2016; Jakobi et al., 2018). The average epithermal neutron count rate across all campaigns was $38\,919 \pm 5\,815$ cph prior to corrections;

All detector systems were set to integrate neutron counts over 10 seconds, and the driving speed ranged from 10 to 100 meters

300 per minute. Within the two months of the field campaign, roving measurements took place on 13 days (see Fig. 6).

### 3.7 Local Neutron Monitor and Bonner Spheres

The neutron intensity at ground level strongly depends on the incoming cosmic-ray neutron flux. In order to account for variations of that incoming flux, researchers typically use neutron monitor (NM) recordings from the Neutron Monitor Database





(http://www.nmdb.eu) as a baseline. For that purpose, Hawdon et al. (2014) and Schrön et al. (2016) recommended to select
a neutron monitor with a cutoff rigidity similar to the study location. Consequently, we suggest neutron monitors Irkutsk and
Jungfraujoch as potential candidates for the reference flux:

- Fendt site, $R_{\text{cut}} = 4.14 \pm 0.02\,\text{GV}$ (calculation based on Herbst et al., 2013), altitude $595\,\text{m}$,

- Irkutsk NM, $R_{\text{cut}} \approx 3.64\,\text{GV}$, altitude $435\,\text{m}$,

- Jungfraujoch NM, $R_{\text{cut}} \approx 4.49\,\text{GV}$, altitude $3570\,\text{m}$.

Nevertheless, open questions about the suitability of neutron monitor data for local CRNS applications still exist, as pointed
out by Schrön (2017) and Baroni et al. (2018). To build the basis for a more thorough analysis in future studies, we take
advantage of additional neutron detector instruments during the study period in Fendt, the so-called Mini-NM (Krüger et al.,
2008), and Bonner Spheres (Bramblett et al., 1960; Rühm et al., 2012). Both instruments were installed close to CRNS probe
#15, and measure different neutron energies due to different thickness of the polyethylen shielding (PE).

The Bonner Spheres are spherical $^{3}$He proportional counters (3.3 cm diameter, type SP9, Centronic Ltd.) surrounded by
polythylene shields of 3, 5, and 9 inch, respectively. An additional bare detector without any surrounding material was used
to get a high response to thermal neutrons. To increase the response to high-energy neutrons ($> 20\,\text{MeV}$), the 9" detector has
been modified by adding a 0.5" lead shell (Mares and Schraube, 1997). Combinations of multiple Bonner spheres can be used
to estimate the spectral flux distribution (i.e., the energy spectrum) of secondary cosmic-ray neutrons (Leuthold et al., 2007).

Figure 5 shows a comparison of the count rates from the remote neutron monitors at Jungfraujoch and Irkutsk, the local
Mini-NM, and the local Bonner Spheres. Future studies will investigate the environmental factors that have an effect on the
various neutron energies and which of the detectors is most suitable for the incoming neutron corrections.

### 3.8 Soil data and local soil moisture observations

As pointed out in section 1.2, the estimation of volumetric soil moisture from epithermal neutron count rates typically requires a
sufficient number of soil moisture and bulk density measurements in the sensor footprint. These measurements commonly orig-
inate from thermo-gravimetry (the retrieval of soil moisture as the mass difference of a soil core before and after oven-drying),
but may as well be acquired with other measurement techniques. The more reference measurements within the footprint are
available, the more reliably the effects of spatial variability of soil moisture can be accounted for in the calibration.

In this study, we combined several measurement techniques. The Rott headwater catchment study site at Fendt is partly
equipped with permanent soil moisture measurement devices (SoilNet, see section 3.8.1, and Fig. 1), which provided long-
term spatially- and temporally dense records. For the observation period, the existing network was temporarily extended by
additional wireless sensors (see 3.8.2) and profile probes (see 3.8.3).

In addition to these continuous observations, manual sampling was carried out in an intensive campaign from June 25–26,
2019 (see section 3.8.4). In that campaign, soil samples were extracted directly at the sensor locations from various depths
for the thermo-gravimetric measurement of water content and other soil properties. Furthermore, vertical profiles of FDR
measurements were carried out at these and additional locations.

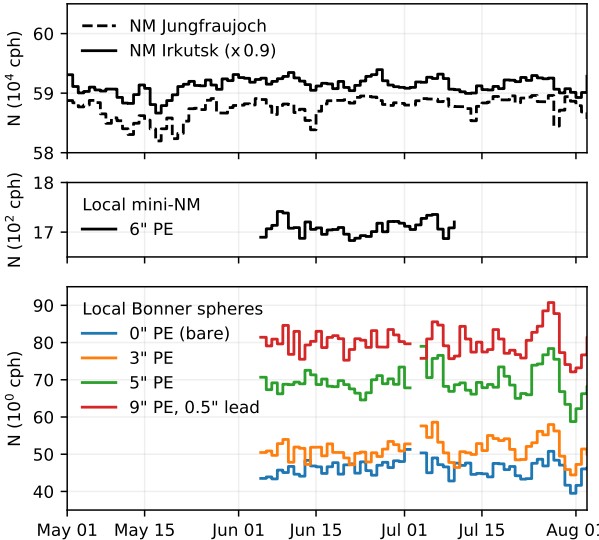

**Figure 5.** Neutron count rate $N$ for different detector types: the remote neutron monitors at Jungfraujoch and Irkutsk, the local Mini-NM, and the local Bonner Spheres. Each instrument is equipped with different PE or lead shields to adjust the detection sensitivity towards different neutron energies.

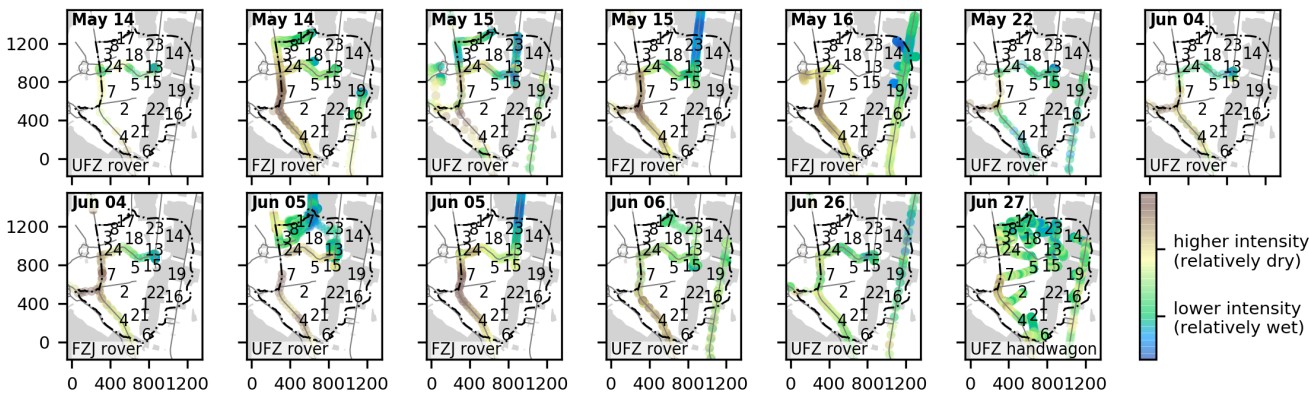

**Figure 6.** Roving tracks and relative neutron intensity. Each panel represents a roving campaign day and a rover type (UFZ rover, FZJ rover, handwagon). The color code qualitatively indicates the variability of observed corrected neutron intensity within a single roving campaign, where a higher neutron intensity corresponds to drier conditions. Forested and built-up areas as well as roads are shown in grey, the watershed is represented by the broken black line; stationary CRNS sensor locations by the sensor's ID. Basemap data from OSM (© OpenStreetMap contributors, 2019), see section 4.4.



### 3.8.1 Permanent soil sensor network (SoilNet)

The permanent soil moisture monitoring data at Fendt are available since June 2015. In total, 55 vertical profiles were distributed in the north-west part of the Rott headwater catchment, covering an area of about 9 ha (Fig. 1). Each profile records permittivity, and temperature in 5, 20, and 50 cm depth, every 15 minutes. The system is an implementation of SoilNet (version 3) - a ready-made system, developed at Forschungszentrum Jülich GmbH, Germany (SoilNet, 2018). Permittivity and temperature are measured redundantly at each depth with two slightly displaced sensors (SMT100, Truebner GmbH, Neustadt, Germany). The SMT100 uses a ring oscillator to determine the permittivity of the soil from the electromagnetic propagation velocity. The relation between sensor counts and permittivity $\epsilon$ was individually calibrated for each device prior to installation according to Bogena et al. (2017) and Qu et al. (2013). We used the dielectric mixing model of Birchak et al. (1974) to link $\epsilon$ with the volumetric water content ($\theta$)

$$\theta = \frac{\epsilon^{0.5} - (1-n)\,\epsilon_s^{0.5} - n\,\epsilon_a^{0.5}}{\epsilon_w^{0.5} - \epsilon_a^{0.5}}. \tag{2}$$

$\epsilon_s$, $\epsilon_a$, and $\epsilon_w$ stand for the permittivities of the solid, air, and water component of the soil. $n$ denotes the porosity of the soil. The permittivity of the air is defined with $\epsilon_a = 1$ and $\epsilon_w$ was computed from temperature $T$ (in °C) according to Weast (1986) with

$$\begin{aligned}
\epsilon_w = 78.54 \cdot \big( & 1 - 4.579 \cdot 10^{-3}\,(T - 25) \\
& + 1.19 \cdot 10^{-5}\,(T - 25)^2 \\
& - 2.8 \cdot 10^{-8}\,(T - 25)^3 \big).
\end{aligned} \tag{3}$$

For the permanent installation area, the values for the solid permittivity $\epsilon_s = 3.29$ and the porosity of the soil $n = 0.63$ were derived from a laboratory analysis of nine representative soil samples, following the methodology proposed by Qu et al. (2016).

We provide the SoilNet data in a compressed binary netCDF-4 format. The 15-minute interval timeseries cover the period May 1$^{\text{st}}$–July 31$^{\text{st}}$. We stored each variable in a separate field with the dimensions *profile ID, depth, and time*. The geographical locations of the profiles are also contained in the file thus enabling geostatistical analyses. We did not fill data gaps that resulted from sensor malfunction or transmission errors.

### 3.8.2 Temporary soil sensor network

In addition to the permanent SoilNet we installed flexible variants of such a sensor system, the so-called „Wireless soil moisture sensor networks" (WSN, see also Schrön et al., 2018; Lausch et al., 2018). The system is an implementation of the BaseNet system (IMMS gGmbH, Ilmenau, Germany), developed in cooperation with the Helmholtz Centre for Environmental Research (UFZ, Leipzig, Germany), and includes standard SMT100 probes (Truebner GmbH, Neustadt, Germany).

Across the catchment, nine locations were equipped with vertical measurement profiles: eight locations with two profiles each, and one location with four profiles (i.e. 18 profiles in total). The locations were chosen to cover different land use types and to closely accompany the CRNS sensors #4, #9, #11, #13, #14 (four profiles), #19, #22, and #24 (see Fig. 1). Soil permittivity and temperature are measured with slightly displaced sensors at 15, 30, and 45 cm depth and in intervals of 10 minutes.



The measuring principle and calibration procedure correspond to the explanations in section 3.8.1. The corresponding porosity
values of the soil were derived from a laboratory analysis of two representative soil samples for 15 cm and 30 cm depth at each
location. For the solid soil components, we assumed a permittivity of $\epsilon_s = 3.29$ following the considerations of section 3.8.1.

### 3.8.3 Soil moisture profile probes

Soil moisture profile probes were installed throughout the study area in order to monitor the vertical distribution of soil moisture
over time. The vertical distribution pattern is not only affecting the vertical CRNS footprint (Köhli et al., 2015), but also allows
to better constrain vertical water movement and thus groundwater recharge (Baroni et al., 2018).

We employed FDR-based profile probes PR2 of three variants (PR2/6 analogue, PR2/4 SDI, PR2/6 SDI, Delta-T Devices
LLC, Cambridge, England, UK). These were installed in the direct vicinity of the 19 stationary CRNS sensors outside the
SoilNet, at a maximum distance of 1.5 m. PR2/4 measures at 10, 20, 30, and 40 cm depth, PR2/6 additionally yields values for
60 and 100 cm depth. Tab. 2 specifies the maximum measurement depths covered at each CRNS location.

The voltage readings $U_{\text{raw}}$ recorded every 20 min were corrected and processed as described in section 3.8.4, using the
manufacturer's equation (Delta-T, 2016)

$$
\begin{aligned}
\sqrt{\epsilon} = {}& 1.125 - 5.53\,U_c + 67.17\,U_c{}^2 - 234.42\,U_c{}^3 \\
& + 413.56\,U_c{}^4 - 356.68\,U_c{}^5 + 121.53\,U_c{}^6
\end{aligned}
\tag{4}
$$

and applying eq. (6) after manually removing spurious data.

### 3.8.4 Manual soil moisture observations and soil sampling

To increase the spatial coverage of the continuously-recorded soil moisture data, to enable the calibration of the sensors and
also to obtain basic soil properties (bulk density, residual water content, organic matter content, texture), a large number of
corresponding measurements were carried out from June 25 to 26, 2019.

The standard method for measuring soil moisture and other soil properties is collecting soil cores from various depths, and
390 analysing them with thermo-gravimetry (referred to as *thermo-gravimetric profiles* in the following). Yet, that approach is also
the most labour-intensive. It was not possible to apply that sampling procedure at a sufficient number of locations within two
days. Therefore, the thermo-gravimetric approach was complemented by FDR measurements at the same depths (referred to
as *FDR profiles* in the following). While manpower limited the total number of sampled sites, access permission posed further
restrictions on their spatial distribution (see Fig. 1).

Altogether, the following sampling design was applied during the intensive campaign: Within 2 m around each CRNS sensor,
a *thermo-gravimetric profile* was collected (details below). In the direct vicinity, an *FDR profile* was collected as an additional
reference (details below). Four FDR profiles, surrounding the CRNS sensor in all cardinal directions within 3–6 m distance,
complemented the survey in the close proximity to the CRNS sensors. Finally, randomly selected locations (under the con-
straints of access permission and of accounting for different land cover types in the footprint) served for closing remaining gaps



in the design. The resulting collection consisted of 23 thermo-gravimetric and 139 FDR profiles. All measurement locations were surveyed with DGPS.

For collecting a *thermo-gravimetric profile*, a pit was excavated. Soil cores were horizontally extracted with cylinders (4 cm height, 5.6 cm diameter), at depths of 0 to 25 cm with an increment of 5 cm, where the measurement depth signified the distance between the soil surface and the upper edge of the sampling ring. Two replicate cores were extracted for each depth in each pit. Water content was determined by drying the samples at a temperature of 105 °C, and subsequent weighing. Analyses of residual water content and organic matter were performed for composite samples from three classes: (1) "forest" on "mineral" soil, (2) "other land use" on "mineral", or (3) "other land use" on "organic" soil. For each class, mixed samples for each 5-cm-increment served for analyses, which consisted of exposing the samples to 400 and 1000 °C for 16 and 12 h, respectively. For the mineral composite samples, a texture analysis was conducted using wet sieving and laser diffraction (Helos, Sympatec GmbH, Germany).

The *FDR profile* resulted from handheld ML2 ThetaProbes (Delta-T Devices LLC, Cambridge, UK) performed in vertically-augered holes of incrementally increasing depths. The depth increments of 5 cm correspond to those of the thermo-gravimetric measurements. Here, the depth refers to the upper end of the electrodes after the probe had been fully inserted. At each depth, the probe was vertically inserted, read, and extracted three times, with a slight rotation after each time, in order to capture micro-scale variability. Sensor voltage $U_{raw}$ served as the primary variable recorded. Since $U_{raw}$ varied considerably between individual sensors, each reading was linearly converted to corrected voltage $U_c$. This conversion was based on sensor-specific calibration performed in air and water. The resulting corrected values of $U_c$ allowed their conversion to permittivity $\epsilon$ according to manufacturer's specifications (Delta-T, 1998):

$$\sqrt{\epsilon} = 1.07 - 6.4 U_c + 6.4 U_c{}^2 - 4.7 U_c{}^3. \tag{5}$$

We tested various published equations for converting $\epsilon$ to the $\theta$ determined from thermo-gravimetry at the points of concomitant measurements. Since none of the tested relations performed satisfactorily, we re-fitted the coefficients of these equations. The best fit was achieved with the re-adjusted equation of Zhao et al. (2016) resulting in

$$\theta = \frac{-0.3846\,\varrho - 1.1037 + \sqrt{\epsilon}}{-2.2595\,\varrho + 1.7629 + (0.8995\,\varrho + 0.6479) \cdot \sqrt{\epsilon}}, \tag{6}$$

with $\varrho$ being the dry bulk density in units of g/cm$^3$. The variable $\varrho$ was related to the $\epsilon$ according to the respective soil stratum and depth layer as described above.

### 3.9 Vegetation and biomass

Cosmic ray neutron sensors are affected by all hydrogen pools within the footprint. Therefore, water stored in plants and hydrogen as a component of the plant tissue had to be quantified. The applied methods differ for grassland (more dynamic due to mowing operations) and forest (higher total biomass).

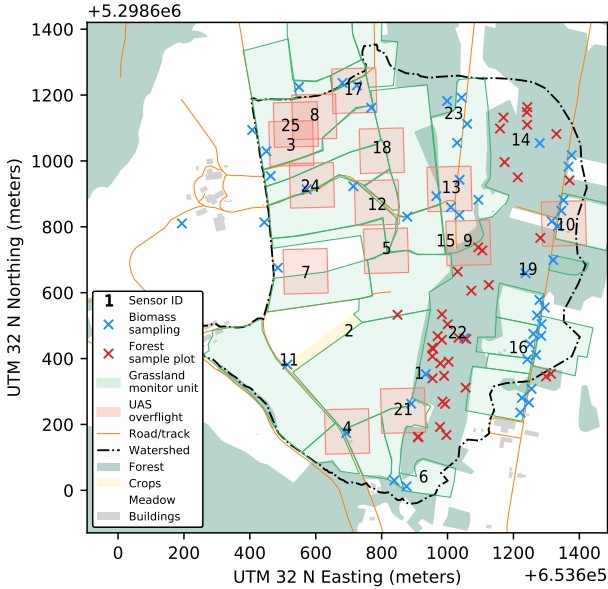

**Figure 7.** The layout of this figure corresponds to Fig. 1, but shows additional observation campaigns, namely the locations of vegetation biomass sampling (blue crosses, see section 3.9.1), areal units for monitoring the status of meadows (light green), forest/tree surveys (red crosses, see section 3.9.2), and UAS overflight zones (light red, see section 3.10). Basemap data from OSM (© OpenStreetMap contributors, 2019), see section 4.4.

### 3.9.1 Biomass on grassland and cropland

Above-ground biomass on grass- and cropland sites was sampled three times (May 14–16, June 6 and July 17) at the same 45 locations (see Fig. 7). For that purpose, all plant material within a $30\,\mathrm{cm} \times 30\,\mathrm{cm}$ frame was harvested and immediately weighed in the field, using a mobile scale. Subsequently, the samples were stored in labelled paper bags for transport to the lab and oven dried to constant weight at $65\,°\mathrm{C}$.

The vegetation water content can be estimated from weight loss after drying. For the amount of hydrogen and oxygen stored in cellulose, Franz et al. (2013) suggested a stoichiometric approximation (55.6 %).

Most grassland patches experienced multiple farming operations during the campaign, namely mowing, drying of the cut grass, baling, and removal of hay. These operations generally took place on patches defined by their respective ownership. The status of these patches (see Fig. 7) was visually inspected every 3–4 days and recorded according to the above-mentioned stages. These records should allow assessing the effects of management on the above-ground biomass dynamics.

### 3.9.2 Biomass in forest

Woodland covers a considerable fraction of the study area. While this forest is largely dominated by spruce (*Picea abies*), it also hosts smaller groves or individual trees of beech (*Fagus sylvatica*), alder (*Alnus glutinosa*), and ash (*Fraxinus excelsior*). The



heterogeneous age of the stands makes it more difficult to estimate forest biomass. The collected data are intended to enable

composite methods for the estimation of biomass employing spectral and LIDAR-based (see 4.3) remote sensing imagery and tree inventory data (e.g., Brovkina et al., 2017).

Forest mapping consisted of a *plot-based* and a *tree-based* survey (for locations, see Fig. 7). The *plot-based* mapping provides ground-truth for species classification. We mapped the forested area during June-August 2019. We selected 29 sites consisting exclusively or largely of one tree species, and recorded species, position (handheld GPS), diameter of the patch and

the approximate stand height (three measurements with laser rangefinder TruPulse 360B, LTI, Centennial, USA). The *tree-based* survey comprised four plots surveyed similarly to the method of Brovkina et al. (2017): within a circular plot of 25 m diameter, we recorded all single trees with at least 7 cm diameter, measuring their azimuth and distance from the centre of the plot, height (TruPulse 360B) and their circumference (tape measure) as a surrogate for breast-height-diameter.

Additionally, the forest undergrowth and litter mass was determined at six locations. For this purpose, all litter and plant

material within a 30 cm × 30 cm frame was collected and processed in the same scheme as the crop- and grassland site samples (i.e. weighing in the field, oven-drying at 65 °C to constant weight). Water equivalents in undergrowth and litter mass were estimated by weight loss after drying summed with the stoichiometric amount of hydrogen and oxygen stored in cellulose (Franz et al., 2013, 55.6 %).

### 3.10    Thermal images from remote sensing

Thermal imagery was acquired covering a 60 m radius of 14 different CRNS sensors (see Fig. 7) as an indicator for soil moisture distribution within the footprint area of highest contribution to the CRNS signal (Köhli et al., 2015). The times and dates of every flight are summarized in Fig. 8.

For the data acquisition the UAS MK Okto XL 6S12 (HiSystems GmbH, Moormerland, Germany) equipped with a radio-metric calibrated FLIR Tau 2 336 (FLIR Systems, Inc., Wilsonville, OR, USA) was used. This thermal camera uses a VOx

microbolometer focal plane array which is sensitive to wavelengths from 7.5–13.5 $\mu$m. Manufacturers specify an accuracy of $\pm 5°$C. The camera was upgraded with an external shutter system (TeAx Technology UG, Wilnsdorf, Germany) to achieve a better radiometric accuracy and avoid the vignetting effect in the individual scenes (personal communication R. Schlepphorst 2019, May 5). Further features are its 9 mm focal length and a sensor resolution of 336 × 256 pixels, leading to a 35° × 27° field of view.

Flights were performed at an altitude of 100 m and a speed of approximately 5 m s$^{-1}$ with a horizontal and vertical scene overlap of 82 % × 84 %. Within each area of interest, five to ten ground control plates were distributed to enable accurate georectifica-tion of the geomatic products. Their positions were determined with a Leica Zeno GG04 (Leica Geosystems AG, Heerbrugg, Switzerland) DGPS antenna with sub-pixel accuracy.

Three steps of data preprocessing were performed before the creation of orthomosaics: For each individual scene capture,

the camera creates multiple frames. The frame with the highest image quality according to the Agisoft Image Quality tool is





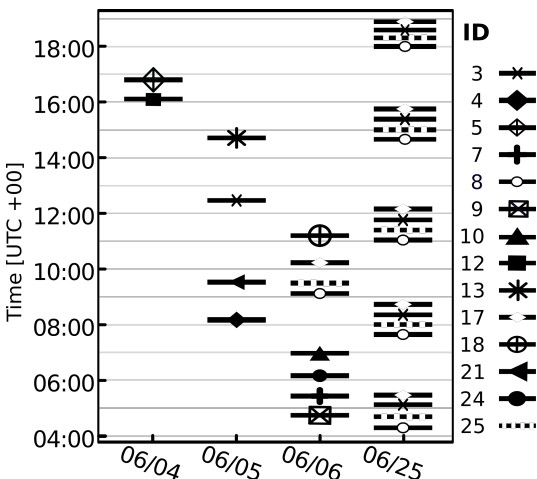

**Figure 8.** Summary of UAS flights. The areas correspond to the CRNS sensor ID (see Fig. 1). Every flight had a flight duration of approximately 10 minutes.

selected for further processing. The extracted frames were temperature corrected as proposed by Maes et al. (2017)

$$T_{\text{UAScorr}} = T_{\text{UAS}} - T_{\text{air}} + T_{\text{air\_mean}}, \tag{7}$$

with $T_{\text{UAScorr}}$ and $T_{\text{UAS}}$ being the corrected and uncorrected measured temperature by the camera, respectively, $T_{\text{air}}$ the air temperature at the moment of image capture and $T_{\text{air\_mean}}$ the mean of the air temperature during the flight. Air temperature of

the meteorological data (see 3.3) served as input for the correction. The orthomosaics for each flight were created in Metashape Professional (Version 1.5.3, Agisoft LLC, St. Petersburg, Russia). The application of a Wallis filter (Wallis, 1977) increased the contrast within the individual frames and thus enabled automatic feature recognition by the image alignment algorithms. The filtered images are again substituted by the temperature corrected frames when building the orthomosaic in 'mosaic' mode. The RMSE of the positions of the ground control plates is included in the metadata of each flight.

**3.11 Discharge**

Discharge observations, which are, for example, needed for setting up hydrological or land surface models of the stuy area, were derived from water level measurements (Datalogger Type 575-II, HT-Hydrotechnik, Obergünzburg, Germany), taken at the outlet of the Rott headwater catchment in the north (see Fig. 1) with ten minutes resolution. The rating curve was derived from salt-tracer–electric conductivity measurements. Due to the lack of discharge reference values for channel water levels

above 40 cm, discharge rates exceeding 0.1 m$^3$ s$^{-1}$ are increasingly uncertain.

**3.12 Groundwater levels**

For hydrological modeling and water balance assessments, information about the groundwater dynamics are also of interest. The saturated zone at the Fendt valley bottom can be differentiated into a shallow aquifer situated on top and between the





Quaternary sediment layers and a thicker, deeper confined aquifer. For each groundwater layer, a hydraulic head measurement

is available during the campaign period. The observation well for the shallow aquifer is located in the vicinity of the climate station, and the one for the deeper aquifer is situated below the country road, south of the SoilNet at the center between the CRNS sensors #12, #18, and #24 (see Fig. 1). The time series were recorded with three hourly (Datalogger Type 575-II, HT-Hydrotechnik, Obergünzburg, Germany) and 15-minute (Hobo U20L-04, Onset, Bourne, MA, USA) resolution for the shallow and deep aquifer, respectively. As the well opening for the shallow aquifer is flush with the ground surface and located in a

small depression, strong precipitation events with ponding can cause a direct water flow into the well tube. Therefore, sharp peaks should be interpreted carefully and the groundwater temperature data should be considered as indicator for such events, too. Negative peaks in the time series are due to local pumping tests.

## 4   Related data from third parties

This section introduces several additional and useful data sets that are not part of this data publication, and are provided by

institutions or research collaborators without direct involvement in the Cosmic Sense project.

### 4.1   Complementary data by ScaleX and MOSES heat wave campaign 2019

The Cosmic Sense joint field campaign 2019 was carried out at the same time as the ScaleX 2019 campaign of KIT Campus Alpin, which involved additionally the MOSES (Modular Observation Solutions for Earth System) test campaign for the heat wave event chain. From these activities, complementary measurements were performed including sensible and latent heat

fluxes, net ecosystem exchange, turbulence statistics, planetary boundary layer depth, surface temperature, surface emissivity, surface thermal infrared images, vertical profiles of wind speed and direction, and water vapor and air temperature LIDAR profiles at Hohenpeißenberg. As of now, access to this data needs to be requested from the individual project leaders of ScaleX 2019. Further description and contact information is available at https://scalex.imk-ifu.kit.edu.

### 4.2   Long-term hydro-meteorological observations

Long term hydrometeorological and ecological data for TERENO Pre-Alpin and the Rott headwater catchment (Fendt) are available via the TERENO Data Discovery Portal (https://www.tereno.net/ddp/). Further details about the TERENO Pre-Alpine observatory are available in Kiese et al. (2018) and via the KIT Campus Alpin's website (https://www.imk-ifu.kit.edu/tereno. php.)

### 4.3   Digital Elevation Model and soil overview maps

A high-resolution digital elevation (DEM) or terrain model (DTM) can be helpful particularly for hydrological applications, e.g., for identifying flow paths or estimating the depth of the groundwater table, or for forest biomass estimation (see section 3.9). DEM and DTM products at various horizontal resolutions and vertical accuracy can be obtained from the correspond-



ing state agency, the Bayerisches Landesamt für Digitalisierung, Breitband und Vermessung (https://www.ldbv.bayern.de), e.g., the DEM1 product with a resolution of 1 m × 1 m, and a vertical accuracy better than 20 cm.

Soil maps (Bodenübersichtskarte 1:25000) for Bavaria are available in vector (shapefile) format published under CC BY-3.0 license by the Bavarian Environmental Agency (LfU, https://www.lfu.bayern.de/umweltdaten/geodatendienste/pretty_downloaddienst.htm?dld=uebk25).

### 4.4    Land use, roads, waterways

During fieldwork and for visualisation, we used OpenStreetMap data layers (OpenStreetMap contributors, 2019) available via
http://download.geofabrik.de, namely landuse, waterways, and traffic. The data are distributed under ODbL license (www.openstreetmap.org/copyright).

### 4.5    Meteorological data

The German Weather Service (DWD) provides open climate data via https://opendata.dwd.de. These include comprehensive open-access observations of climate variables at the climate station Hohenpeißenberg (identifier 2290, 977 m ASL) which is
located about five kilometres south-west of the study area.

## 5    Example application

The prime motivation of this paper is to present a comprehensive data set to the scientific community. That data set provides all the information required to estimate, analyse and put into context spatio-temporal soil moisture patterns from different sensors, and at different scales.

### 5.1    The stationary CRNS network

The heart of the data set is the dense cluster of CRNS sensors. Although an in-depth data analysis is, by definition, beyond the scope of this data publication, we would still like to give an impression of the potential of this methodology as well as of the temporal and spatial variability of soil moisture in the context of this study. In order to convey such an impression, we applied a standard processing workflow to estimate soil moisture from neutron count rates. As we only consider this an "illustration",
not a study result, we only briefly outline the corresponding steps in the following:

– **Standardize** neutron count rates to a common sensitivity level, which is the sensitivity of the calibrator sensor (see Tab. 2);

– **Correct** neutron count rates for the effects of incoming cosmic neutron flux, barometric pressure, and atmospheric water vapor. For that purpose, we applied the standard procedure summarized by Andreasen et al. (2017) in the section on
"Correcting Neutron Intensity". To correct for the effects of pressure and water vapor, we used data from the climate



gauge (see section 3.3) for all CRNS sensors; to correct for incoming neutron flux, we used the data from the neutron monitor at Jungfraujoch (see section 3.7);

– **Calibrate** the $N_0$ parameter from Eq. (1) for each CRNS sensor based on the soil moisture measurements obtained in the intensive manual sampling campaign (see 3.8.4). For computing the average observed soil moisture in a sensor footprint, we use the iterative horizontal and vertical weighting procedure suggested by Schrön et al. (2017);

– **Average** (or smooth) neutron count rates in time in order to increase the signal-to-noise ratio. In order to illustrate the behaviour over the entire study period, we applied a moving average with a window size of 24 hours; convert the smoothed neutron intensity to volumetric soil moisture using the calibrated $N_0$ and Eq. (1);

– **Interpolate** the soil moisture estimates $\theta_{\text{sensor},jk}$ obtained from each CRNS sensor $j$ at each time step $k$ to values at the pixel scale $\theta_{\text{pixel}}$. For that purpose, we constructed a $10\,\text{m} \times 10\,\text{m}$ grid inside the spatial bounding box of the catchment. For each grid pixel $i$, a weight $w_{ijk}$ was inferred for each CRNS sensor $j$ from the value of the approximated horizontal weighting function as provided by Schrön et al. (2017), Eq. B1. Then, the soil moisture $\theta_{\text{pixel},ik}$ at each pixel $i$ was computed as a weighted average based on Eq. (8):

$$\theta_{\text{pixel},ik} = \frac{\sum w_{ijk}\theta_{\text{sensor},jk}}{\sum w_{ijk}}. \tag{8}$$

It should be noted that the above processing workflow uses only parts of the entire data set in order to roughly characterise soil moisture patterns in space and time. The scientific potential of the data set, however, is in combining the various observations at various scales. Yet, Fig. 9 already illustrates the pronounced dynamics of soil moisture as well as the spatial variability, i.e. within the catchment area and over the campaign duration. From May 20 to 22, an unusually intense and persistent rainfall event resulted in more than 125 mm of rainfall in less than 48 hours, followed by another 40 mm towards the end of May. That sequence of events led to saturated conditions in the second half of May. Over June and July, the catchment was subject to substantial drying, interrupted by occasional, but intense rainfall events. The median volumetric soil moisture dropped from a maximum of 65 % to a minimum of 37 %. Over the same period, volumetric soil moisture always strongly varied in space: At the driest period around early July, the wettest five percent of the catchment still exceeded a soil moisture of 45 %. The wettest parts of the catchment are located around CRNS sensor #23 (see Fig. 1 and 2) and are characterised by peat soils and very shallow groundwater tables.

## 5.2 Spatial patterns using mobile CRNS

The mobile roving campaign on June 27 (Fig. 6 and Fig. 10) brings the spatial patterns to a higher level of detail. The spatial distribution of soil moisture in the Rott headwater catchment is already visible from the CRNS network, but only with the mobile measurements is it possible to cross the fields between the stationary sensors and to significantly extend spatial coverage and resolution. The campaign was conducted with the UFZ rover on a hand wagon and the data has been aggregated on a regular

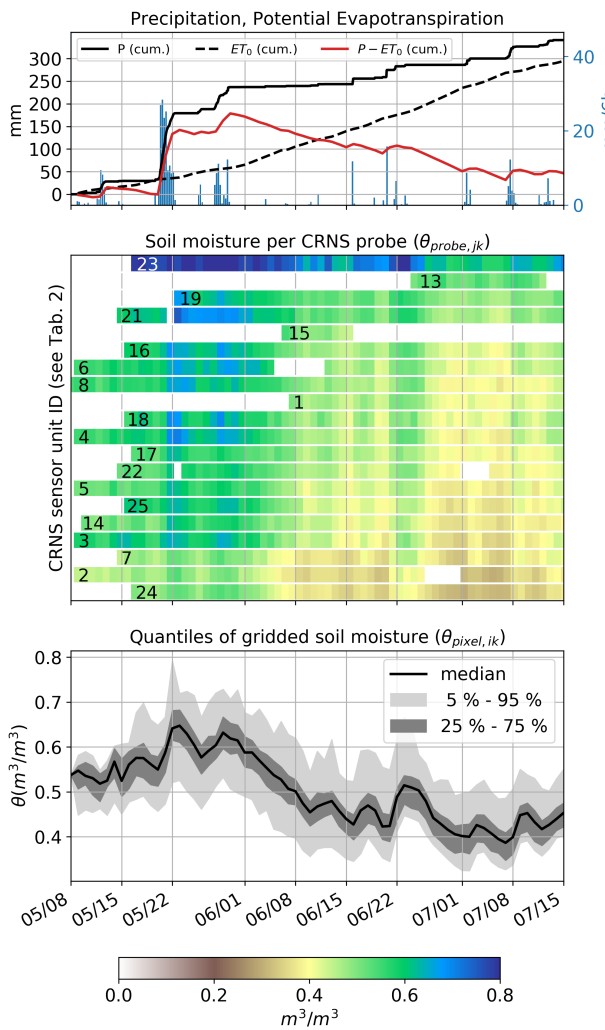

**Figure 9.** Dynamics of meteorological variables and soil moisture as estimated from neutron count rates. Upper panel: cumulative precipitation P, reference evapotranspiration $ET_0$ (FAO, 1998) based on Penman-Monteith, and the difference $P - ET_0$. Center panel: soil moisture $\theta_{\mathrm{sensor},jk}$ estimated from neutron count rates for each of the CRNS sensors, where the sensors have been sorted from bottom to top in ascending order based on average soil moisture - the white spaces indicate periods of missing data; please note that the sensors #9–#12 are not included in this overview because they did not record valid data during the time of collocation with the calibrator sensor or the manual sampling campaign. Bottom panel: Temporal dynamics of different soil moisture quantiles after the soil moisture estimates have been interpolated to a grid (see main text to explanation).

30 m grid. Neutron data has undergone basic atmospheric corrections (see section 4.5), corrections for soil properties using the SoilGrids data base (Hengl et al., 2017; Fersch et al., 2018) and a reduction by 10 % at locations governed by forest-type land use in order to account for the biomass effect (Baatz et al., 2015). The overall wetness gradient in the area has been confirmed,





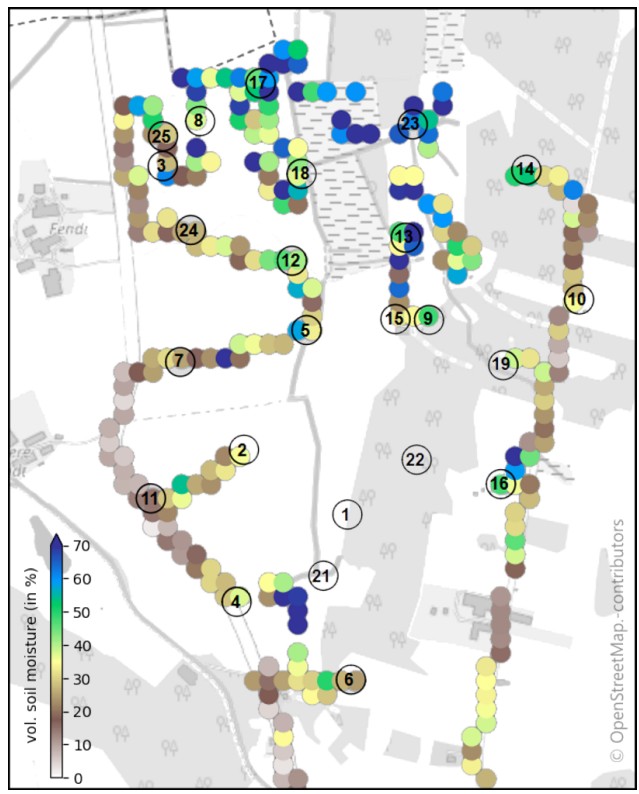

**Figure 10.** Roving with a hand wagon, passing every stationary sensor except #1, #21 and #22. Basemap © OpenStreetMap contributors 2020. Distributed under a Creative Commons BY-SA License.

but the mobile measurements revealed a much larger wet spot in the north, which could not have been resolved by the stationary sensors due to a limited number of sensors located in that area. Besides the wet soil in the peat area and despite the downward shielding of the roving sensor, the observations are still clearly influenced by the lower water content of the roads which are typically slightly elevated over the surrounding areas (see Schrön et al., 2018, for a description of this effect). In prospective analyses of the data for the determination of field soil moisture, it will be necessary to correct for this ‚road effect‘, e.g., by applying the procedure developed by Schrön et al. (2018).

## 6   Conclusions

With this study, we present and provide a unique and comprehensive data set to several research communities: to those who investigate methods to retrieve soil moisture from cosmic ray neutron sensing, to those who study the variability of soil moisture at different spatio-temporal scales, and to those who intend to better understand the role of root-zone soil moisture dynamics in the context of catchment and groundwater hydrology, as well as land-atmosphere exchange processes.





The data set is *unique* in that it involves, for the first time, a dense network of 24 CRNS sensors in a catchment area of 1 km$^2$. That catchment exhibited pronounced moisture gradients from wet peat to much drier loamy soils, and the campaign period was characterised by strong wetting and drying dynamics.

The data set is *comprehensive* in that it not only includes all the data that is required to interpret the neutron count rates from stationary sensors (meteorological time series, local soil moisture observations, soil properties such as bulk density and
organic carbon content, and vegetation biomass), but also various rich data sets that allow to put CRNS-based soil moisture estimates into various spatio-temporal contexts: CRNS roving campaigns, thermal imaging from multiple UAS overflights, multiple WSN clusters, as well as time series of discharge observations at the catchment outlet and groundwater levels.

In this way, the presented data set will be a valuable resource to those seeking a better understanding of cosmic ray neutron signals and for advancing scientific tools for CRNS-based soil moisture retrieval, and to those who aim to use these data and
instruments for hydrological and hydrogeological applications.

## 7   Code and data availability

For data sharing and archiving we used the EUDAT Collaborative Data Infrastructure (https://eudat.eu). This pan-European e-infrastructure offers numerous data-related services ranging from sharing, archiving, processing to publication of the data. Its realization is closely related to the European Open Science Cloud (EOSC), a European initiative aimed at promoting
free sharing of scientific data. In the context of this data publication, this project benefited from the services B2SHARE and B2HANDLE, both targeted at sharing, publishing and guaranteeing long-term persistence of data and managing persistent identifiers.

The data described in this paper are available from EUDAT (https://doi.org/10.23728/b2share.85fe0f9dac0f48df9215c17e65d1f1e1, Fersch et al., 2020a) [*Remark: These preliminary URIs are valid during the reviewing of this manuscript. They may need to*
*be updated after the revision to a permanent URI*]. For more fine-grained access to the large files from the thermal imagery, these are hosted under a separate DOI (https://doi.org/10.23728/b2share.93ed99e486904d48a8a6a68083066198, Fersch et al., 2020b) . The repository structure corresponds to the subsections of section 3 (see also Tab. 3). Besides the data, each folder contains a JSON-file (readable ASCII text) holding meta data and additional format descriptions, where these differ from the conventions described in section 3.2.
We developed the R-package *FDR2soilmoisture*, ver. 0.107 (Francke, 2019) for processing the FDR-data, and made it freely available.

*Author contributions.*  The lead authors Benjamin Fersch, Till Francke, Maik Heistermann, Martin Schrön, Veronika Döpper and Jannis Jacobi were in charge of designing and conducting the JFC, processed the data and drafted this manuscript. Michael Förster, Sascha Oswald, Heye Bogena, Andreas Güntner, Harald Kunstmann and Steffen Zacharias are PIs of *Cosmic Sense* and participated in the fieldwork. Gabriele
Baroni, Theresa Blume, Harrie-Jan Hendricks-Franssen, Birgit Kleinschmit, Ulrich Schmidt are further PIs of the group who contributed to the planning of the campaign. Mandy Kasner, Amol Patil, Daniel Rasche, Lena Scheiffele, Jannis Weimar designed and conducted field-



**Table 3.** Overview of data categories and their path in the repository.

| Section | Variable | Path in repository |
|---|---|---|
| 3.3 | meteo data | meteo.zip |
| 3.4 | CRNS-stationary | crns_stationary.zip/ |
| 3.6 | CRNS-roving | crns_roving.zip |
| 3.7 | Neutron Monitors, Mini-NM, Bonner Spheres | crns_incoming.zip |
| 3.8.1 | soil moisture, permanent WSN[*] | soilmoisture_permanent_wsn.zip |
| 3.8.1 | soil moisture, temporary WSN[*] | soilmoisture_temporary_wsn.zip |
| 3.8.3 | soil moisture, profile probes | soilmoisture_profile_probes.zip |
| 3.8.4 | soil properties, soil moisture, manual sampling | soilmoisture_manual_sampling.zip |
| 3.9 | vegetation / biomass | biomass.zip |
| 3.10 | thermal imagery | (part II, separate DOI) |
| 3.11 | discharge | discharge.zip |
| 3.12 | groundwater | groundwater.zip |

[*] Wireless soil moisture sensor network

work as members of *Cosmic Sense*, Markus Köhli supported the data analysis. Tobias Gränzig conducted fieldwork and contributed to data processing; Christian Budach conducted a large share of the fieldwork; Sandra Szulc-Seyfried coordinated the campaign and participated in the fieldwork. Marek Zreda inspected the field site and served as an advisor. All authors contributed to the writing of the manuscript.
Bernd Heber provided the Mini-NM and contributed to the analysis of its data. Ralf Kiese contributed to the conducting and analysis of the gravimetric soil sampling at Fendt. Vladimir Mares installed the Bonner Spheres and processed the data. Hannes Mollenhauer developed and installed the temporary wireless sensor networks and carried out the corresponding data processing. Ingo Völksch maintained the permanent soil moisture network at Fendt and processed the sensor data for publication together with Benjamin Fersch.

*Competing interests.* J. Weimar and M. Köhli hold CEO positions at StyX Neutronica GmbH, Heidelberg, Germany. M. Zreda has been
scientific adviser of Lab C, LLC, Sheridan, USA.



*Acknowledgements.* This research was funded by the Deutsche Forschungsgemeinschaft (DFG, German Research Foundation) – project 357874777 of the research unit FOR 2694 "Cosmic Sense". The TERrestrial Environmental Observatory (TERENO) Pre-Alpine infrastructure is funded by the Helmholtz Association and the Federal Ministry of Education and Research. We thank the Scientific Team of the ScaleX Campaign 2019 for their contribution. The joint field campaign was also supported by the MOSES project (Modular Observation Solutions

for Earth Systems) of the Helmholtz Association which also funded the Jülich CRN rover. We further express out gratitude to Konstantin Herbst (Uni Kiel) for calculating the cutoff rigidity of the Fendt site, and Thomas Brall (HMGU) and Florian Wagner (HMGU) for their support during the installation of the Bonner Spheres. We would like to thank Paterzell airfield control for their cooperation and permission to use their airspace for UAS based data acquisition. We are indebted to all landowners granting permission to access their property and enduring additional heavy traffic during the already painful restrictions by the road construction works. The tremendous efforts in the field

was only possible thanks to the help of the technical staff and involved students. We gratefully acknowledge the services provided by EUDAT (namely B2DROP, B2SHARE, B2HANDLE), which greatly facilitated the workflows within the project and the publication of these data. Base map data copyrighted OpenStreetMap contributors and available from https://www.openstreetmap.org.



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
