# Peer review of "A dense network of cosmic-ray neutron sensors for soil moisture observation in a pre-alpine headwater catchment in Germany"

_Earth System Science Data, 2020_

## Referee Comment (RC1) · Anonymous Referee #1 · 16 Apr 2020

General comments

This paper describes a number of extensive data sets collected concurrently from a headwater catchment in Germany. The availability of such a diverse range of measurements in a geographically restricted area offers unique opportunities for CRNS data analysis and testing of data correction and modelling approaches.

The authors are to be commended for collating such a high quality dataset.

A number of grammatical corrections are required (see specific comments below) but the paper is generally well written.

[Figure]

I am not sure if the title of the paper is a good representation of the content as there is just as much data on meteorology and alternative soil moisture information as there is on CRNS measurements. Not sure how best to capture the breadth of observations but perhaps a title along the lines of the following might be better...

A dense network of cosmic-ray neutron sensors for soil moisture observation in a pre-alpine headwater catchment in Germany including supplementary meteorological and hydrological observations

OR

A dense network of cosmic-ray neutron sensors and supplementary measurements for soil moisture observation in a pre-alpine headwater catchment in Germany

The datasets are well organised with suitable metadata and data structures used throughout. I would like to have seen a version of the CRNS static data with an estimate of soil moisture data using a predefined approach. I realise techniques are evolving and the key focus is on CRNS processing options but this would provide an off-the-shelf option for some users who just want some measure of soil moisture for analysis (i.e. hydrologic model, satellite cal/val). This may increase data utilisation. As it stands the raw counts and all of the data required for corrections and processing are provided but users with little experience in the CRNS field may be a little reluctant to process the data themselves.

Specific comments

Line 8 – change 'that network' to 'this network' Line 10 – change to '... allow users to ...' Line 19 – change to '...EU-DAT and is split into two subsets..." Line 19 – 1st hyperlink includes first part of reference and does not work Line 29 – maybe spell out TDR and TDR first time through Line 31 – do you mean support volume Line 36 – add 'are' between 'and' and 'thus' Line 39 – it might be less clunky to just say they have poor temporal resolution Line 44 – not sure describing the footprint as exponentially

shaped is a good description. The footprint declines radially with distance from the CRNS – its more bell shaped or declines exponentially with distance from the CRNS Line 48 – '. . .presented the first. . .' Line 58 – change to ". . . monitoring exists and these use various gases. . ." Line 87 – suggest deleting after the : Line 88 – needs rewording – does not make sense as is Line 96 – change 'of' to 'in' Line 130 – change 'had' to 'have' Line 149 – delete 'however' Line 156 – by backbone do you mean existing long-term datasets? If so suggest chaning Line 160 – change 'could identify' to 'identified' Line 170 – insert 'this was' between 'and' and 'followed' Line 213 – do you mean back ground? Figure 2 is not referenced in the text Fig3 caption – use "Note: y-axis scaling differs between plots" Line 361 & Line 362 – inverted commas wrong Line 374 – needs to be reworded for clarity. What is being constrained? Models? Para starting Line 386 – change to "A large number of manual soil moisture measurements were carried out from June 25 to 26, 2019 and these were useful for increasing the spatial coverage of the continuously-recorded soil moisture data, enabling the calibration of the moisture sensors and obtaining basic soil properties (bulk density, residual water content, organic matter content, texture). Line 475 – change 'is' to 'was' Line 493 – 'on top of' Line 567 – remove 'yet' and 'already' Line 578 – delete 'already' Line 600 – change to ". . .allow users to. . ."
* * *

---

## Referee Comment (RC2) · Anonymous Referee #2 · 7 May 2020

The work presented by Fersch and others introduces new combined dataset for a small catchment of about 1 sqkm area combining a dense network of cosmic-ray soil mois-ture sensors with additional hydrometeorological data. The dataset presented relates to a field experiment campaign in 2019 that lasted for almost three months covering late-spring and part of the summer in Germany. The manuscript is well-written and presented with good quality figures. There is no question about the volume of data collected and its quality as the group of scientists is very strong in this area. I believe this is a very good set of data to investigate soil moisture dynamics in a 1-sq-km area for climate conditions ad soil/vegetation characteristics similar to this site. I have very few comments:

[Figure]

(1) I think the context chosen by the authors to justify the choice of area is slightly incomplete (or maybe slightly misleading). Reading the relevant sections of the manuscript, it gives the impression that this choice was because typical applications of cosmic-ray sensors are either done for single (isolated) sites or for large-scale continental/national networks, which consequently misses this overlapping aspect of multiple measurements at the same 1 sq-km (grid) area (note that the area represents a tiny fraction of the 55 sq-km Rott catchment). The nature of single sites of large networks is continuous monitoring capability whereas here the data describes a short-term summer campaign (from late-April to early-July). There is nothing majorly wrong with the text but given that this is short campaign, framing the paper more towards the importance of short-term intensive campaigns would, in my opinion, make the paper present better the motivation for this dataset. Authors can cite similar work done for other products such as SMOSMANIA and SMAPEx campaigns, for example. In addition, an area of 1 sq-km is probably related to very small catchments, and the context would probably be better linked to understanding spatial heterogeneity for high-resolution soil moisture applications. I am surprised to see the authors failed to cite seminal work by Roger Grayson, Andrew Western, and Gunther Bloschl in the 1990s that tackle this high-resolution measurement problem with very interesting and impactful results to the hydrological community.

(2) In many instances throughout the paper, I don't understand why the manuscript jumps to figures without following a common numerical order (introduces Figures 1 previously, then refers to Figure 9 here; then refers to Fig 3 later on; Fig 2 is not referred in the text before showing up in the manuscript, etc...). This makes reading through the manuscript a bit more difficult.

(3) The cosmic-ray neutron sensing technique for soil moisture application is now more than 10 years old. I'd strongly encourage the authors and the community, in general, to stop using words like "emerging", "novel", "promising", etc... to describe the technique as these statements now read a bit outdated

(4) L70-71: Perhaps the authors can include the citation to previous studies that have altered the values of the 'a' coefficients in an attempt to improve the estimated soil moisture signal from the sensor:

Rivera Villarreyes, C. A., Baroni, G., & Oswald, S. E. (2011). Integral quantification of seasonal soil moisture changes in farmland by cosmic-ray neutrons. Hydrology and Earth System Sciences, 15(12), 3843–3859. https://doi.org/10.5194/hess-15-3843-2011

Iwema, J., Rosolem, R., Baatz, R., Wagener, T., and Bogena, H. R.: Investigating temporal field sampling strategies for site-specific calibration of three soil moisture–neutron intensity parameterisation methods, Hydrol. Earth Syst. Sci., 19, 3203–3216, https://doi.org/10.5194/hess-19-3203-2015, 2015.

Heidbüchel, I., Güntner, A., and Blume, T.: Use of cosmic-ray neutron sensors for soil moisture monitoring in forests, Hydrol. Earth Syst. Sci., 20, 1269–1288, https://doi.org/10.5194/hess-20-1269-2016, 2016.

(5) Figure 6: There is just too much information in this figure. What is the main purpose of the figure? Do the authors need to show all numbers on those maps? DO they need to show all maps?

(6) As an example of application, the authors could easily have identified how similar or different the measurements are (20+ sensors in a 1 sq-km area seems a bit too much for me). Perhaps they could either look at the different CDF for each measurement and determine if they theoretically come from the same or different distribution using the Kolmogorov-Smirnov test; or apply some temporal stability analysis (???)
* * *

---

## Author Comment (AC1) · 15 Jun 2020

**Author Response to Reviews of**

**A dense network of cosmic-ray neutron sensors for soil moisture observation in a pre-alpine headwater catchment in Germany**

Benjamin Fersch et al.

*Earth Syst. Sci. Data Discuss.,* `doi:10.5194/essd-2020-48`
* * *
**RC:** *Reviewer Comment*,     AR: *Author Response*,     ☐ Manuscript text

Dear Referees,

we would like to thank you very much for your positive comments and constructive suggestions to our manuscript. In this document, we would like to provide our responses to the comments of each of the referees in one single document and to outline the corresponding changes to the manuscript. We hope that our response together with the revision of the manuscript sufficiently addresses the referees' concerns.

Kind regards,

Benjamin Fersch

on behalf of the author team

**1. Referee #1**

**1.1. Title**

> A dense network of cosmic-ray neutron sensors for soil moisture observation in a pre-alpine headwater catchment in Germany

**RC:** *I am not sure if the title of the paper is a good representation of the content as there is just as much data on meteorology and alternative soil moisture information as there is on CRNS measurements. Not sure how best to capture the breadth of observations but perhaps a title along the lines of the following might be better...*

"A dense network of cosmic-ray neutron sensors for soil moisture observation in a pre-alpine headwater catchment in Germany including supplementary meteorological and hydrological observations"

*or*

"A dense network of cosmic-ray neutron sensors and supplementary measurements for soil moisture observation in a pre-alpine headwater catchment in Germany"

AR: *We thank the referee for the suggestion. We would like to propose the following changed title: "A dense*

network of cosmic-ray neutron sensors for soil moisture observation in a highly instrumented pre-alpine headwater catchment in Germany".

**1.2. Availability of a soil moisture product**

**RC:** *The datasets are well organised with suitable metadata and data structures used throughout. I would like to have seen a version of the CRNS static data with an estimate of soil moisture data using a predefined approach. I realise techniques are evolving and the key focus is on CRNS processing options, but this would provide an off-the-shelf option for some users who just want some measure of soil moisture for analysis (i.e. hydrological model, satellite cal/val). This may increase data utilisation. As it stands the raw counts and all of the data required for corrections and processing are provided, but users with little experience in the CRNS field may be a little reluctant to process the data themselves.*

AR: *We certainly understand the referee's concern that the presented data would not be helpful for users with little experience. Yet, the referee pointed out himself that the retrieval of a soil moisture product from neutron count rates is still evolving and not trivial. The aim of this data publication is to provide a data set to the research community that can be used for soil moisture estimation at varying levels of complexity. Adding any preliminary or "first-order" soil moisture product as part of the published data set would not be a valuable service to the community. Providing an in-depth analysis, on the other hand, is certainly beyond the scope of this publication. Instead, we are currently working on an extensive study that explores options for soil moisture estimation at the level of individual probes, but, more importantly, on the consistent interpolation of soil moisture across the study area. As part of that ongoing study, we also aim at providing a soil moisture product which will then be adequately documented and validated as part of the corresponding publication.*

**1.3. Line 8**

**RC:** *Change 'that network' to 'this network'*

AR: *Will be changed.*

**1.4. Line 10**

**RC:** *change to '...allow users to...'*

AR: *Will be changed.*

**1.5. Line 19**

**RC:** *change to '...EU-DAT and is split into two subsets...'*

AR: *Will be changed to: "EUDAT Collaborative Data Infrastructure (EUDAT), and is split into two subsets:"*

**1.6. Line 19**

**RC:** *1st hyperlink includes first part of reference and does not work*

AR: *We are sorry that the hyperlink did not appear to work for the referee. We double-checked it with different pdf reader formats and it worked as expected. We would like to ask the reviewer to double check the link as well, and we hope for a final solution of the issue during the process of typesetting.*

**1.7. Line 29**

**RC:** *maybe spell out TDR and TDR first time through*

AR: *We assume the referee meant "FDR and TDR". We will spell out both at first time use.*

**1.8. Line 31**

**RC:** *do you mean support volume*

AR: *Correct. We will change "support" to "support volume".*

**1.9. Line 36**

**RC:** *add 'are' between 'and' and 'thus*

AR: *Will be added.*

**1.10. Line 39**

**RC:** *It might be less clunky to just say they have poor temporal resolution.*

AR: *We thank the referee, but we would prefer to keep the sentence as it is since such datasets do not actually have a temporal "resolution", but could just be snapshots at arbitrary points in time.*

**1.11. Line 44**

**RC:** *Not sure describing the footprint as exponentially shaped is a good description. The footprint declines radially with distance from the CRNS – its more bell shaped or declines exponentially with distance from the CRNS.*

AR: *Thank you for pointing this out, we have indeed described the shape inappropriately. The shape is actually very complex and not easy to describe in short. Since those details are not relevant here, we think that the provided reference is sufficient and we will remove the term "exponentially-shaped" from the text.*

**1.12. Line 48**

**RC:** *'...presented the first...'*

AR: *Will be changed.*

**1.13. Line 58**

**RC:** *change to "...monitoring exists and these use various gases..."*

AR: *Will be changed.*

**1.14. Line 87**

**RC:** *suggest deleting after the :*

AR: *We understand the suggestion. Yet, we prefer to keep the two items after the column as it helps guiding the reader through what follows next. We propose to replace the colon with "such as".*

**1.15.  Line 88**

**RC:** *needs rewording - does not make sense as is*

AR: *Thank you, we propose to change the sentence to: "The term 'Roving CRNS' stands for the utilization of mobile CRNS sensors to increase the spatial extent of soil moisture measurements. It has been recognized ..."*

**1.16.  Line 96**

**RC:** *change 'of' to 'in'*

AR: *Will be changed.*

**1.17.  Line 130**

**RC:** *change 'had' to 'have'*

AR: *Will be changed.*

**1.18.  Line 149**

**RC:** *delete 'however'*

AR: *Will be deleted.*

**1.19.  Line 156**

**RC:** *By backbone do you mean existing long-term datasets? If so suggest changing.*

AR: *We will change "backbone" to "long-term".*

**1.20.  Line 160**

**RC:** *change 'could identify' to 'identified'*

AR: *Will be changed.*

**1.21.  Line 170**

**RC:** *insert 'this was' between 'and' and 'followed'*

AR: *Will be added.*

**1.22.  Line 213**

**RC:** *Do you mean back ground?*

AR: *We will remove the term "backbone".*

**1.23.  Figure 2**

**RC:** *Figure 2 is not referenced in the text.*

AR: *We propose to change the text as follows: "Tab. 2 gives an overview of all stationary CRNS sensors that were used during the campaign. Fig. 2 illustrates the proportions of the different sensor models."*

**1.24. Fig3 caption**

RC: *Use "Note: y-axis scaling differs between plots"*

AR: *Will be changed.*

**1.25. Line 361 Line 362**

RC: *inverted commas wrong*

AR: *Will be corrected.*

**1.26. Line 374**

RC: *Needs to be reworded for clarity. What is being constrained? Models?*

AR: *We suggest to replace "but also allows to better constrain" by "it also informs about".*

**1.27. Line 386**

RC: *change to "A large number of manual soil moisture measurements were carried out from June 25 to 26, 2019 and these were useful for increasing the spatial coverage of the continuously-recorded soil moisture data, enabling the calibration of the moisture sensors and obtaining basic soil properties (bulk density, residual water content, organic matter content, texture).*

AR: *Will be adopted.*

**1.28. Line 475**

RC: *change 'is' to 'was*

AR: *Will be changed.*

**1.29. Line 493**

RC: *'on top of'*

AR: *Will be changed.*

**1.30. Line 567**

RC: *remove 'yet' and 'already'*

AR: *Will be removed.*

**1.31. Line 578**

RC: *delete 'already'*

AR:   *Will be deleted.*

**1.32. Line 600**

RC:   **change to "...allow users to..."**

AR:   *Will be changed.*

**2. Reviewer #2**

**2.1. Introduction**

RC:   ***I think the context chosen by the authors to justify the choice of area is slightly incomplete (or maybe slightly misleading). Reading the relevant sections of the manuscript, it gives the impression that this choice was because typical applications of cosmic-ray sensors are either done for single (isolated) sites or for large-scale continental/national networks, which consequently misses this overlapping aspect of multiple measurements at the same 1 sq-km (grid) area (note that the area represents a tiny fraction of the 55 sq-km Rott catchment). The nature of single sites of large networks is continuous monitoring capability whereas here the data describes a short-term summer campaign (from late-April to early-July). There is nothing majorly wrong with the text but given that this is short campaign, framing the paper more towards the importance of short-term intensive campaigns would, in my opinion, make the paper present better the motivation for this dataset. Authors can cite similar work done for other products such as SMOSMANIA and SMAPEx campaigns, for example. In addition, an area of 1 sq-km is probably related to very small catchments, and the context would probably be better linked to understanding spatial heterogeneity for high-resolution soil moisture applications. I am surprised to see the authors failed to cite seminal work by Roger Grayson, Andrew Western, and Gunther Bloschl in the 1990s that tackle this high-resolution measurement problem with very interesting and impactful results to the hydrological community.***

AR:   *We thank the referee for this thoughtful comment. Yet, we tend to disagree with the view that the prime feature of the presented campaign was its short-term character – in comparison to long-term national scale networks. At least we interpret the following comment accordingly: "[...] given that this is a short campaign, framing the paper more towards the importance of short-term intensive campaigns would, in my opinion, make the paper present better the motivation for this dataset." In our view, however, the prime feature of this campaign was the high spatial density of CRNS sensors in a study area characterised by large heterogeneity of land use, soil, and soil moisture. That spatial density is motivated, on the one hand, by the aim to cover root zone soil moisture dynamics at the small catchment scale, and, on the other hand, by the aim to represent the spatial variability of soil moisture at that scale, and to use the overlap of CRNS footprints for that purpose. As soon as you place CRNS probes so far from each other that their footprints do not overlap anymore, the retrieval of spatial patterns is "reduced" to the problem of spatial interpolation. However, as the footprints overlap, there is the potential to extract additional spatial information from the fact that the same area is covered by different probes (similar to the idea of a geophysical inversion). We have tried to explain these aspects in ll. 121–125 of the original manuscript.*

      *So while we appreciate the comment, we think that the short-term character of this campaign is not its prime feature nor its motivation. The duration was rather constrained by resources. However, we have already started to establish a dense network of 12 CRNS sensors near Potsdam that is designed at least for a medium-term operation (i.e. several years).*

*Based on the above argument, we would also prefer not to explicitly cite the suggested references on SMOSMANIA and SMAPEX. While SMAPEX was specifically geared towards understanding soil moisture variability inside the SMAP footprint scale (i.e. 10–30 km), SMOSMANIA is about soil moisture monitoring at 21 locations as part of the national scale meteorological observation network RADOME (Meteo France). SMOS and SMAP themselves are already referred to in the overview papers we cited in section 1.1 (Wang and Qu, 2009; Mohanty et al., 2017). Yet, we thank the referee for suggesting Blöschl and Grayson who of course cannot be cited too often. In the revised version, we will refer to "Blöschl, G., R. Grayson (2000): Spatial Observations and Interpolation. In: Blöschl, G., R. Grayson (Eds.): Spatial Patterns in Catchment Hydrology - Observations and Modelling, Cambridge University Press, pp. 17-50" in l. 30 of the original manuscript.*

**2.2.  Figure numbering**

**RC:** *In many instances throughout the paper, I don't understand why the manuscript jumps to figures without following a common numerical order (introduces Figures 1 previously, then refers to Figure 9 here; then refers to Fig 3 later on; Fig 2 is not referred in the text before showing up in the manuscript, etc...). This makes reading through the manuscript a bit more difficult.*

 AR: *We apologize for any inconvenience, but using the Copernicus tex template, we could not influence the actual placement of the figures in the manuscript draft. We will take additional care of this issue during the copy-editing phase. As for Fig. 2, we will add the reference. Fig. 9 constitutes an exception, as it is used both for illustrating the temporal coverage and presenting example. Because the focus is on the latter, it is placed closer to this sections.*

**2.3.  Is the CRNS technique still "novel"?**

**RC:** *The cosmic-ray neutron sensing technique for soil moisture application is now more than 10 years old. I'd strongly encourage the authors and the community, in general, to stop using words like "emerging", "novel", "promising", etc... to describe the technique as these statements now read a bit outdated.*

 AR: *We agree with your view that the CRNS method is used for various applications and that since its broader introduction in 2010 a certain standard has emerged. On the other hand, we don't think that the technique has yet evolved as a standard for soil moisture observation per se. We will go through the text and adapt our wording while acknowledging the current spread and history of the technique.*

**2.4.  Lines 70-71**

**RC:** *Perhaps the authors can include the citation to previous studies that have altered the values of the 'a' coefficients in an attempt to improve the estimated soil moisture signal from the sensor:*

Rivera Villarreyes, C. A., Baroni, G., & Oswald, S. E. (2011). Integral quantification of seasonal soil moisture changes in farmland by cosmic-ray neutrons. Hydrol. Earth Syst. Sci., 15(12), 3843–3859. https://doi.org/10.5194/hess-15-3843-2011

Iwema, J., Rosolem, R., Baatz, R., Wagener, T., and Bogena, H. R.: Investigating temporal field sampling strategies for site-specific calibration of three soil moisture–neutron intensity parameterisation methods, Hydrol. Earth Syst. Sci., 19, 3203–3216, https://doi.org/10.5194/hess-19-3203-2015, 2015.

Heidbüchel, I., Güntner, A., and Blume, T.: Use of cosmic-ray neutron sensors for soil moisture monitoring in forests, Hydrol. Earth Syst. Sci., 20, 1269–1288,https://doi.org/10.5194/hess-20-1269-2016, 2016.

AR:    *Thank you for suggesting additional papers about the $a_i$ parameters. We propose to reformulate the sentence: "The parameters $a_0$, $a_1$, and $a_2$ have proved to be robust in their original formulation across multiple CRNS sites (Desilets2010, Evans2016, Schroen2017, among others), while few locations apparently work better with different values (Rivera2011, Iwema2015, Heidbuchel2016).*

**2.5.   Figure 6**

RC:   ***There is just too much information in this figure. What is the main purpose of the figure? Do the authors need to show all numbers on those maps? Do they need to show all maps?***

AR:    *Figure 6 shows the tracks and patterns of the roving campaign. We will replace the IDs of the CRNS-probes with simple symbols for clarity. However, we would like to show the maps for all dates to illustrate the spatiotemporal coverage of the rover campaigns and the resulting qualitative spatial patterns of soil moisture.*

**2.6.   Example application**

RC:   ***As an example of application, the authors could easily have identified how similar or different the measurements are (20+ sensors in a 1 sq-km area seems a bit too much for me). Perhaps they could either look at the different CDF for each measurement and determine if they theoretically come from the same or different distribution using the Kolmogorov-Smirnov test; or apply some temporal stability analysis (???)***

AR:    *We thank the referee for this interesting suggestion, which we will try to incorporate in our ongoing analysis of the data. Without doubt, there are countless other analyses that could be performed with the presented data. Therefore, we are happy to provide these data to the community. However, we cannot and do not intend to present any exhaustive analysis in this paper. In accordance to the ESSD journal guidelines, we are focusing on the documentation of the dataset only. Our presented application is merely one of many examples of how this dataset could be used.*

---

## Author Response (AR1)

**Author Response to Reviews of**

**A dense network of cosmic-ray neutron sensors for soil moisture observation in a pre-alpine headwater catchment in Germany**

Benjamin Fersch et al.
*Earth Syst. Sci. Data Discuss.,* `doi:10.5194/essd-2020-48`
* * *
RC: *Reviewer Comment*,     AR: *Author Response*,     ☐ Manuscript text

Dear Referees,

we would like to thank you very much for your positive comments and constructive suggestions to our manuscript. Here, we compiled our responses for the comments of each of the referees into one single document and outlined the corresponding changes made to the manuscript. We hope that our response together with the revision of the manuscript sufficiently addresses the referees' concerns.

Kind regards,

Benjamin Fersch

on behalf of the author team

**1. Referee #1**

**1.1. Title**

> A dense network of cosmic-ray neutron sensors for soil moisture observation in a pre-alpine headwater catchment in Germany

RC: *I am not sure if the title of the paper is a good representation of the content as there is just as much data on meteorology and alternative soil moisture information as there is on CRNS measurements. Not sure how best to capture the breadth of observations but perhaps a title along the lines of the following might be better...*

"A dense network of cosmic-ray neutron sensors for soil moisture observation in a pre-alpine headwater catchment in Germany including supplementary meteorological and hydrological observations"

 *or*

"A dense network of cosmic-ray neutron sensors and supplementary measurements for soil moisture observation in a pre-alpine headwater catchment in Germany"

AR:   *We thank the referee for the suggestion. We would like to propose the following changed title: "A dense network of cosmic-ray neutron sensors for soil moisture observation in a highly instrumented pre-alpine headwater catchment in Germany".*

**1.2.  Availability of a soil moisture product**

RC:   ***The datasets are well organised with suitable metadata and data structures used throughout. I would like to have seen a version of the CRNS static data with an estimate of soil moisture data using a predefined approach. I realise techniques are evolving and the key focus is on CRNS processing options, but this would provide an off-the-shelf option for some users who just want some measure of soil moisture for analysis (i.e. hydrological model, satellite cal/val). This may increase data utilisation. As it stands the raw counts and all of the data required for corrections and processing are provided, but users with little experience in the CRNS field may be a little reluctant to process the data themselves.***

AR:   *We certainly understand the referee's concern that the presented data would not be helpful for users with little experience. Yet, the referee pointed out himself that the retrieval of a soil moisture product from neutron count rates is still evolving and not trivial. The aim of this data publication is to provide a data set to the research community that can be used for soil moisture estimation at varying levels of complexity. Adding any preliminary or "first-order" soil moisture product as part of the published data set would not be a valuable service to the community. Providing an in-depth analysis, on the other hand, is certainly beyond the scope of this publication. Instead, we are currently working on an extensive study that explores options for soil moisture estimation at the level of individual probes, but, more importantly, on the consistent interpolation of soil moisture across the study area. As part of that ongoing study, we also aim at providing a soil moisture product which will then be adequately documented and validated as part of the corresponding publication.*

**1.3.  Line 8**

RC:   ***Change 'that network' to 'this network'***

AR:   *Changed.*

**1.4.  Line 10**

RC:   ***change to '...allow users to...'***

AR:   *Changed.*

**1.5.  Line 19**

RC:   ***change to '...EU-DAT and is split into two subsets...'***

AR:   *Changed to: "EUDAT Collaborative Data Infrastructure (EUDAT), and is split into two subsets:"*

**1.6.  Line 19**

RC:   ***1st hyperlink includes first part of reference and does not work***

AR:   *We are sorry that the hyperlink did not appear to work for the referee. We double-checked it with different pdf reader formats and it worked as expected. We would like to ask the reviewer to double check the link as well, and we hope for a final solution of the issue during the process of typesetting.*

**1.7. Line 29**

**RC:** *maybe spell out TDR and TDR first time through*

AR: *We assume the referee meant "FDR and TDR". We spelled out both at first time use in the revised version.*

**1.8. Line 31**

**RC:** *do you mean support volume*

AR: *Correct. We changed "support" to "support volume".*

**1.9. Line 36**

**RC:** *add 'are' between 'and' and 'thus*

AR: *Added.*

**1.10. Line 39**

**RC:** *It might be less clunky to just say they have poor temporal resolution.*

AR: *We thank the referee, but we would prefer to keep the sentence as it is since such datasets do not actually have a temporal "resolution", but could just be snapshots at arbitrary points in time. Thank you, we changed "infrequent measurement intervals" to "poor temporal resolution".*

**1.11. Line 44**

**RC:** *Not sure describing the footprint as exponentially shaped is a good description. The footprint declines radially with distance from the CRNS – its more bell shaped or declines exponentially with distance from the CRNS.*

AR: *Thank you for pointing this out, we have indeed described the shape inappropriately. The shape is actually very complex and not easy to describe in short. Since those details are not relevant here, we think that the provided reference is sufficient and we removed the term "exponentially-shaped" from the text.*

**1.12. Line 48**

**RC:** *'...presented the first...'*

AR: *Changed.*

**1.13. Line 58**

**RC:** *change to "...monitoring exists and these use various gases..."*

AR: *Changed.*

**1.14. Line 87**

**RC:** *suggest deleting after the :*

AR: *We understand the suggestion. Yet, we prefer to keep the two items after the column as it helps guiding the*

*reader through what follows next. We replaced the colon with "such as".*

**1.15. Line 88**

**RC:** *needs rewording - does not make sense as is*

AR: *Thank you, we changed the sentence:*

> The term *Roving CRNS*  stands for the utilization of mobile CRNS sensors  to increase the spatial extent of soil moisture measurements. .

**1.16. Line 96**

**RC:** *change 'of' to 'in'*

AR: *Changed.*

**1.17. Line 130**

**RC:** *change 'had' to 'have'*

AR: *Changed.*

**1.18. Line 149**

**RC:** *delete 'however'*

AR: *Deleted.*

**1.19. Line 156**

**RC:** *By backbone do you mean existing long-term datasets? If so suggest changing.*

AR: *Changed "backbone" to "long-term".*

**1.20. Line 160**

**RC:** *change 'could identify' to 'identified'*

AR: *Changed.*

**1.21. Line 170**

**RC:** *insert 'this was' between 'and' and 'followed'*

AR: *Added.*

**1.22. Line 213**

**RC:** *Do you mean back ground?*

AR:  *We removed the word "backbone".*

**1.23. Figure 2**

RC:  *Figure 2 is not referenced in the text.*

AR:  *We changed the text as follows: "Tab. 2 gives an overview of all stationary CRNS sensors that were used during the campaign. Fig. 2 illustrates the proportions of the different sensor models."*

**1.24. Fig. 3 caption**

RC:  *Use "Note: y-axis scaling differs between plots"*

AR:  *Changed.*

**1.25. Line 361 & Line 362**

RC:  *inverted commas wrong*

AR:  *Corrected.*

**1.26. Line 374**

RC:  *Needs to be reworded for clarity. What is being constrained? Models?*

AR:  *We suggest to replace "but also allows to better constrain" by "it also informs about":*

> The vertical distribution pattern is not only affecting the vertical CRNS footprint (Köhli et al., 2015), but  it also informs about vertical water movement and thus groundwater recharge (Baroni et al., 2018).

**1.27. Line 386**

RC:  *change to "A large number of manual soil moisture measurements were carried out from June 25 to 26, 2019 and these were useful for increasing the spatial coverage of the continuously-recorded soil moisture data, enabling the calibration of the moisture sensors and obtaining basic soil properties (bulk density, residual water content, organic matter content, texture).*

AR:  *Adopted.*

**1.28. Line 475**

RC:  *change 'is' to 'was*

AR:  *Changed.*

**1.29. Line 493**

RC:  *'on top of'*

AR:  *Changed.*

**1.30.  Line 567**

 **RC:**  *remove 'yet' and 'already'*

 AR:  *Removed.*

**1.31.  Line 578**

 **RC:**  *delete 'already'*

 AR:  *Deleted.*

**1.32.  Line 600**

 **RC:**  *change to "...allow users to..."*

 AR:  *Changed.*

**2. Referee #2**

**2.1. Introduction**

**RC:** *I think the context chosen by the authors to justify the choice of area is slightly incomplete (or maybe slightly misleading). Reading the relevant sections of the manuscript, it gives the impression that this choice was because typical applications of cosmic-ray sensors are either done for single (isolated) sites or for large-scale continental/national networks, which consequently misses this overlapping aspect of multiple measurements at the same 1 sq-km (grid) area (note that the area represents a tiny fraction of the 55 sq-km Rott catchment). The nature of single sites of large networks is continuous monitoring capability whereas here the data describes a short-term summer campaign (from late-April to early-July). There is nothing majorly wrong with the text but given that this is short campaign, framing the paper more towards the importance of short-term intensive campaigns would, in my opinion, make the paper present better the motivation for this dataset. Authors can cite similar work done for other products such as SMOSMANIA and SMAPEx campaigns, for example. In addition, an area of 1 sq-km is probably related to very small catchments, and the context would probably be better linked to understanding spatial heterogeneity for high-resolution soil moisture applications. I am surprised to see the authors failed to cite seminal work by Roger Grayson, Andrew Western, and Gunther Bloschl in the 1990s that tackle this high-resolution measurement problem with very interesting and impactful results to the hydrological community.*

**AR:** *We thank the referee for this thoughtful comment. Yet, we tend to disagree with the view that the prime feature of the presented campaign was its short-term character - in comparison to long-term national scale networks. At least we interpret the following comment accordingly: "[...] given that this is a short campaign, framing the paper more towards the importance of short-term intensive campaigns would, in my opinion, make the paper present better the motivation for this dataset." In our view, however, the prime feature of this campaign was the high spatial density of CRNS sensors in a study area characterised by large heterogeneity of land use, soil, and soil moisture. That spatial density is motivated, on the one hand, by the aim to cover root zone soil moisture dynamics at the small catchment scale, and, on the other hand, by the aim to represent the spatial variability of soil moisture at that scale, and to use the overlap of CRNS footprints for that purpose. As soon as you place CRNS probes so far from each other that their footprints do not overlap anymore, the retrieval of spatial patterns is "reduced" to the problem of spatial interpolation. However, as the footprints overlap, there is the potential to extract additional spatial information from the fact that the same area is covered by different probes (similar to the idea of a geophysical inversion). We have tried to explain these aspects in ll. 121–125 of the original manuscript.*

*So while we appreciate the comment, we think that the short-term character of this campaign is not its prime feature nor its motivation. The duration was rather constrained by resources. However, we have already started to establish a dense network of 12 CRNS sensors near Potsdam that is designed at least for a medium-term operation (i.e. several years).*

*Based on the above argument, we would also prefer not to explicitly cite the suggested references on SMOSMANIA and SMAPEX. While SMAPEX was specifically geared towards understanding soil moisture variability inside the SMAP footprint scale (i.e. 10–30 km), SMOSMANIA is about soil moisture monitoring at 21 locations as part of the national scale meteorological observation network RADOME (Meteo France). SMOS and SMAP themselves are already referred to in the overview papers we cited in section 1.1 (Wang and Qu, 2009; Mohanty et al., 2017). Yet, we thank the referee for suggesting Blöschl and Grayson who of course cannot be cited too often. In the revised version, we refer to "Blöschl, G., R. Grayson (2000): Spatial Observations and Interpolation. In: Blöschl, G., R. Grayson (Eds.): Spatial Patterns in Catchment Hydrology*

*- Observations and Modelling, Cambridge University Press, pp. 17-50" in l. 30 of the original manuscript.*

**2.2. Figure numbering**

**RC:** *In many instances throughout the paper, I don't understand why the manuscript jumps to figures without following a common numerical order (introduces Figures 1 previously, then refers to Figure 9 here; then refers to Fig 3 later on; Fig 2 is not referred in the text before showing up in the manuscript, etc...). This makes reading through the manuscript a bit more difficult.*

AR: *We apologize for any inconvenience, but using the Copernicus tex template, we could not influence the actual placement of the figures in the manuscript draft. We will take additional care of this issue during the copy-editing phase. As for Fig. 2, we have now referenced it in line 229. Fig. 9 constitutes an exception, as it is used both for illustrating the temporal coverage and presenting example. Because the focus is on the latter, it is placed closer to this sections.*

**2.3. Is the CRNS technique still "novel"?**

**RC:** *The cosmic-ray neutron sensing technique for soil moisture application is now more than 10 years old. I'd strongly encourage the authors and the community, in general, to stop using words like "emerging", "novel", "promising", etc... to describe the technique as these statements now read a bit outdated.*

AR: *We agree with your view that the CRNS method is being used for various applications and that – since its broader introduction in 2010 – it became a certain standard in environmental monitoring. On the other hand, there are still many open questions in the setup and processing of the measurements, compared to, e.g., TDR technology. Moreover, the full potential of CRNS applications has not yet been exploited, such as, e.g., large scale coverage. In the text we have now adapted the wording to better acknowledge the current situation and successful history of the technique.*

**2.4. Lines 70-71**

**RC:** *Perhaps the authors can include the citation to previous studies that have altered the values of the 'a' coefficients in an attempt to improve the estimated soil moisture signal from the sensor:*

Rivera Villarreyes, C. A., Baroni, G., & Oswald, S. E. (2011). Integral quantification of seasonal soil moisture changes in farmland by cosmic-ray neutrons. Hydrol. Earth Syst. Sci., 15(12), 3843–3859. https://doi.org/10.5194/hess-15-3843-2011

Iwema, J., Rosolem, R., Baatz, R., Wagener, T., and Bogena, H. R.: Investigating temporal field sampling strategies for site-specific calibration of three soil moisture–neutron intensity parameterisation methods, Hydrol. Earth Syst. Sci., 19, 3203–3216, https://doi.org/10.5194/hess-19-3203-2015, 2015.

Heidbüchel, I., Güntner, A., and Blume, T.: Use of cosmic-ray neutron sensors for soil moisture monitoring in forests, Hydrol. Earth Syst. Sci., 20, 1269–1288,https://doi.org/10.5194/hess-20-1269-2016, 2016.

AR: *Thank you for suggesting additional papers about the $a_i$ parameters. We reformulated the sentence:*

> The  parameters $a_0$, $a_1$, and $a_2$  have proved to be robust in  their original formulation across multiple CRNS sites (Desilets et al., 2010; Evans et al., 2016; Schrön et al., 2017, among others), while few locations apparently work better with different values (Rivera Villarreyes et al., 2011; Iwema et al., 2015; Heidbüchel et al., 2016) .

**2.5. Figure 6**

**RC:** *There is just too much information in this figure. What is the main purpose of the figure? Do the authors need to show all numbers on those maps? Do they need to show all maps?*

AR: *Figure 6 shows the tracks and patterns of the roving campaign. We have replaced the IDs of the CRNS-probes with simple symbols for clarity. However, we intend to show the maps for all dates to illustrate the spatiotemporal coverage of the rover campaigns and the resulting qualitative spatial patterns of soil moisture.*

**2.6. Example application**

**RC:** *As an example of application, the authors could easily have identified how similar or different the measurements are (20+ sensors in a 1 sq-km area seems a bit too much for me). Perhaps they could either look at the different CDF for each measurement and determine if they theoretically come from the same or different distribution using the Kolmogorov-Smirnov test; or apply some temporal stability analysis (???)*

AR: *We thank the referee for this interesting suggestion, which we will try to incorporate in our ongoing analysis of the data. Without doubt, there are countless other analyses that could be performed with the presented data. Therefore, we are happy to provide these data to the community. However, we cannot and do not intend to present any exhaustive analysis in this paper. In accordance to the ESSD journal guidelines, we are focusing on the documentation of the dataset only. Our presented application is merely one of many examples of how this dataset could be used.*

**3. General Remarks**

- In the submitted version of the data and the manuscript, we provided measurements of four different Bonner Spheres from June to July. However, in July four additional Bonner Spheres were installed (8 in total) which we would like to add also to the revised version of the manuscript. The additional data could help to even better resolve the energy spectrum of the cosmic-ray neutrons at the site. We updated Fig. 5 and the corresponding text accordingly.

- The dataset on EUDAT for part I has been updated an is now available under the DOI `https://doi.org/10.23728/b2share.282675586fb94f44ab2fd09da0856883`. Accordingly, although unchanged, the part II dataset is now available from the DOI `https://doi.org/10.23728/b2share.bd89f066c26a4507ad654e994153358b`

**A dense network of cosmic-ray neutron sensors for soil moisture observation in a highly instrumented pre-alpine headwater catchment in Germany**

Benjamin Fersch[1], Till Francke[2], Maik Heistermann[2], Martin Schrön[3], Veronika Döpper[4], Jannis Jakobi[5], Gabriele Baroni[2,6], Theresa Blume[7], Heye Bogena[5], Christian Budach[2], Tobias Gränzig[4], Michael Förster[4], Andreas Güntner[7,2], Harrie-Jan Hendricks-Franssen[5], Mandy Kasner[3], Markus Köhli[8,9], Birgit Kleinschmit[4], Harald Kunstmann[1,10], Amol Patil[10], Daniel Rasche[7,2], Lena Scheiffele[2], Ulrich Schmidt[8], Sandra Szulc-Seyfried[2], Jannis Weimar[8], Steffen Zacharias[3], Marek Zreda[11], Bernd Heber[12], Ralf Kiese[1], Vladimir Mares[13], Hannes Mollenhauer[3], Ingo Völksch[1], and Sascha Oswald[2]

[1]Karlsruhe Institute of Technology, Campus Alpin (IMK-IFU), Kreuzeckbahnstraße 19, 82467 Garmisch-Partenkirchen, Germany

[2]Institute of Environmental Science and Geography, University of Potsdam, Karl-Liebknecht-Straße 24–25, 14476 Potsdam, Germany

[3]UFZ – Helmholtz Centre for Environmental Research GmbH, Dep. Monitoring and Exploration Technologies, Permoserstr. 15, 04318, Leipzig, Germany

[4]Technical University of Berlin, Geoinformation for Environmental Planning Lab, Straße des 17. Juni 135, 10623 Berlin, Germany

[5]Agrosphere IBG-3, Forschungszentrum Jülich GmbH, Leo-Brandt-Straße, 52425 Jülich, Germany

[6]Department of Agricultural and Food Sciences, University of Bologna, Viale Fanin 50, 40127 Bologna, Italy

[7]GFZ German Research Centre for Geosciences, Section Hydrology, Telegrafenberg, 14473 Potsdam, Germany

[8]Physikalisches Institut, Heidelberg University, Im Neuenheimer Feld 226, 69120 Heidelberg, Germany

[9]Physikalisches Institut, University of Bonn, Nussallee 12, 53115 Bonn, Germany

[10]Institute of Geography, University of Augsburg, Alter Postweg 118, 86159 Augsburg, Germany

[11]Department of Hydrology and Water Resources, University of Arizona, 1133 E. James E. Rogers Way, 85721-0011 Tucson, Arizona, USA

[12]Institute of Experimental and Applied Physics, University of Kiel, Leibnizstraße 11, 24118 Kiel, Germany

[13]Helmholtz Zentrum München, Institute of Radiation Medicine, Ingolstädter Landstraße 1, 85764 Neuherberg, Germany

**Correspondence:** Benjamin Fersch (fersch@kit.edu)

**Abstract.**

[revised manuscript text omitted]

Mares, V. and Schraube, H.: High energy neutron spectrometry with Bonner spheres, in: Proceedings of the IRPA-Symposium, Radiation Protection in Neighbouring Countries of Central Eur. Prague, Czech Republic, pp. 8–12, http://www.iaea.org/inis/collection/NCLCollectionStore/_Public/30/031/30031609.pdf?r=1, 1997.

McJannet, D., Hawdon, A., Baker, B., Renzullo, L., and Searle, R.: Multiscale soil moisture estimates using static and roving cosmic-ray soil moisture sensors, Hydrology and Earth System Sciences, 21, 6049–6067, https://doi.org/10.5194/hess-21-6049-2017, 2017.

Mohanty, B. P., Cosh, M. H., Lakshmi, V., and Montzka, C.: Soil Moisture Remote Sensing: State-of-the-Science, Vadose Zone Journal, 16, vzj2016.10.0105, https://doi.org/10.2136/vzj2016.10.0105, 2017.

Montzka, C., Bogena, H. R., Zreda, M., Monerris, A., Morrison, R., Muddu, S., and Vereecken, H.: Validation of Spaceborne and Modelled Surface Soil Moisture Products with Cosmic-Ray Neutron Probes, Remote Sensing, 9, 103, https://doi.org/10.3390/rs9020103, 2017.

Ochsner, T. E., Cosh, M. H., Cuenca, R. H., Dorigo, W. A., Draper, C. S., Hagimoto, Y., Kerr, Y. H., Larson, K. M., Njoku, E. G., Small, E. E., and Zreda, M.: State of the Art in Large-Scale Soil Moisture Monitoring, Soil Science Society of America Journal, 77, 1888–1919, https://doi.org/10.2136/sssaj2013.03.0093, 2013.

OpenStreetMap contributors: Planet dump retrieved from https://planet.osm.org , https://www.openstreetmap.org, last access: 22-06-2020., 2019.

Qu, W., Bogena, H., Huisman, J., and Vereecken, H.: Calibration of a Novel Low-Cost Soil Water Content Sensor Based on a Ring Oscillator, Vadose Zone Journal, 12, vzj2012.0139, 1–10, https://doi.org/10.2136/vzj2012.0139, 2013.

775 Qu, W., Bogena, H. R., Huisman, J. A., Schmidt, M., Kunkel, R., Weuthen, A., Schiedung, H., Schilling, B., Sorg, J., and Vereecken, H.: The integrated water balance and soil data set of the Rollesbroich hydrological observatory, Earth System Science Data, 8, 517–529, https://doi.org/10.5194/essd-8-517-2016, 2016.

Rivera Villarreyes, C. A., Baroni, G., and Oswald, S. E.: Integral quantification of seasonal soil moisture changes in farmland by cosmic-ray neutrons, Hydrology and Earth System Sciences, 15, 3843–3859, https://doi.org/10.5194/hess-15-3843-2011, 2011.

[revised manuscript text omitted]